# DynFrs: An Efficient Framework for Machine Unlearning in Random Forest

**Shurong Wang[1], Zhuoyang Shen[1], Xinbao Qiao[1], Tongning Zhang[1] & Meng Zhang[1]** *

[1]ZJUI Institute, Zhejiang University
{shurong.22, mengzhang}@intl.zju.edu.cn

## Abstract

Random Forests are widely recognized for establishing efficacy in classification and regression tasks, standing out in various domains such as medical diagnosis, finance, and personalized recommendations. These domains, however, are inherently sensitive to privacy concerns, as personal and confidential data are involved. With increasing demand for *the right to be forgotten*, particularly under regulations such as GDPR and CCPA, the ability to perform machine unlearning has become crucial for Random Forests. However, insufficient attention was paid to this topic, and existing approaches face difficulties in being applied to real-world scenarios. Addressing this gap, we propose the DynFrs framework designed to enable efficient machine unlearning in Random Forests while preserving predictive accuracy. DynFrs leverages subsampling method $\text{occ}(q)$ and a lazy tag strategy LZY, and is still adaptable to any Random Forest variant. In essence, $\text{occ}(q)$ ensures that each sample in the training set occurs only in a proportion of trees so that the impact of deleting samples is limited, and LZY delays the reconstruction of a subtree until demanded, thereby avoiding unnecessary modifications on tree structures. In experiments, applying DynFrs on Extremely Randomized Trees yields substantial improvements, achieving orders of magnitude faster unlearning performance and better predictive accuracy than existing machine unlearning methods for tree-based models. Our code is available here.

## 1 Introduction

Machine unlearning is an emerging paradigm of removing specific training samples from a trained model as if they had never been included in the training set (Cao and Yang, 2015). This concept emerged as a response to growing concerns over personal data security, especially in light of regulations such as the General Data Protection Regulation (GDPR) and the California Consumer Privacy Act (CCPA). These legislations demand data holders to erase all traces of private users' data upon request, safeguarding *the right to be forgotten*. However, unlearning a single data point in most machine learning models is more complicated than deleting it from the database because the influence of any training samples is embedded across countless parameters and decision boundaries within the model. Retraining the model from scratch on the reduced dataset can achieve the desired objective, but is computationally expensive, making it impractical for real-world applications. Thus, the ability of models to efficiently "unlearn" training samples has become increasingly crucial for ensuring compliance with privacy regulations while maintaining predictive accuracy.

Over the past few years, several approaches to machine unlearning have been proposed, particularly focusing on models such as neural networks (Mehta et al., 2022; Cheng et al., 2023), support vector machines (Cauwenberghs and Poggio, 2000), and $k$-nearest neighbors (Schelter et al., 2023). However, despite the progress made in these areas, machine unlearning in Random Forests (RFs) has received insufficient attention. Random Forests, due to their ensemble nature and the unique tree structure, present unique challenges for unlearning that cannot be addressed by techniques developed for neural networks (Bourtoule et al., 2021) and methods dealing with loss functions and gradient (Qiao et al., 2024) which RFs lack. This gap is significant, given that RFs are widely used in critical, privacy-sensitive fields such as medical record analysis (Alam et al., 2019), financial

---

* Corresponding author.

market prediction (Basak et al., 2019), and recommendation systems (Zhang and Min, 2016) for its effectiveness in classification and regression.

To this end, we study an efficient machine unlearning framework dubbed DYNFRS for RFs. One of its components, the OCC($q$) subsampling technique, limits the impact of each data sample to a small portion of trees while maintaining similar or better predictive accuracy through ensemble. DYNFRS resolves three kinds of requests on RFs: to predict the result of the query, to remove samples (machine unlearning), and to add samples (continual learning ) to the model. The highly interpretable data structure of Decision Trees allows us to make the following two key observations on optimizing online (require instant response) RF (un)learning. (1) Requests that logically modify the tree structure (e.g., sample addition and removal) can be partitioned, coalesced, and lazily applied up to a later querying request. (2) Although fully applying a modifying request on a tree might have to retrain an entire subtree, a querying request after those modifications can only observe the updates on a single tree path in the said subtree. Therefore, we can amortize the full update cost on a subtree into multiple later queries that observe the relevant portion (see Fig. 1).

To this effect, we propose the lazy tag mechanism LZY for fine-grain tracking of pending updates on tree nodes to implement those optimizations, which provides a low latency online (un)learning RF interface that automatically and implicitly finds the optimal internal batching strategy within nodes.

We summarize the key contributions of this work in the following:

- **Subsampling**: We propose a subsampling method OCC($q$) that guarantees a $1/q$ times (where $q < 1$) training speedup and an expected $1/q^2$ times unlearning speedup compared to naïve retraining approach. Empirical results show that OCC($q$) brings improvements to predictive performance for many datasets.

- **Lazy Tag**: We introduce the lazy tag strategy LZY that avoid subtree retraining for unlearning in RFs. The lazy tag interacts with modification and querying requests to obtain the best internal batching strategy for each tree node when handling online real-time requests.

- **Experimental Evaluation**: DYNFRS yields a 4000 to 1500000 times speedup relative to the naïve retraining approach and is orders of magnitude faster than existing methods in sequential unlearning and multiple times faster in batch unlearning. In the online mixed data stream settings, DYNFRS achieves an averaged 0.12 ms latency for modification requests and 1.3 ms latency for querying requests on a large-scale dataset.

## 2 RELATED WORKS

Machine unlearning concerns the complicated task of removing specific training sample from a well-trained model (Cao and Yang, 2015). Retraining the model from scratch ensures the complete removal of the sample's impact, but it is computationally expensive and impractical, especially when the removal request occurs frequently. Studies have explored unlearning methods for support vector machines (Cauwenberghs and Poggio, 2000), and $k$-nearest neighbor (Schelter et al., 2023). Lately, SISA (Bourtoule et al., 2021) has emerged as a universal unlearning approach for neural networks. SISA partitions the training set into multiple subsets and trains a model for each subset (sharding), and the prediction comes from the aggregated result from all models. Then, the unlearning is accomplished by retraining the model on the subset containing the requested sample, and a slicing technique is applied in each shard for further improvements. However, as stated in the paper, SISA meets difficulties when applying slicing to tree-based models.

Schelter et al. (2021) introduced the first unlearning model for RFs based on Extremely Randomized Trees (ERTs), and used a robustness quantification factor to search for robust splits, with which the structure of the tree node will not change under a fixed amount of unlearning requests, while for non-robust splits, a subtree variant is maintained for switching during the unlearning process. However, HedgeCut only supports removal of a small fraction (0.1%) of the entire dataset. Brophy and Lowd (2021) introduced DaRE, an RF variant similar to ERTs, using random splits and caching to enhance unlearning efficiency. Random splits in the upper tree layers help preserve the structure, though they would decrease predictive accuracy. DaRE further caches the split statistics, resulting in less subtree retraining. Although DaRE and HedgeCut provide a certain speedup for unlearning, they are incapable of batch unlearning (unlearn multiple samples simultaneously in one request).

Laterly, unlearning frameworks like OnlineBoosting (Lin et al., 2023) and DeltaBoosting (Wu et al., 2023) are proposed, specifically designed for GBDTs, which differ significantly from RFs in training mechanisms. OnlineBoosting adjusts trees by using incremental calculation updates in split gains and derivatives, offering faster batch unlearning than DaRE and HedgeCut. However, it remains an approximate method that cannot fully eliminate the influence of deleted data, and its high computational cost for unlearning individual instances limits its practical use in real-world applications. In the literature, Sun et al. (2023) attempted to lazily unlearn samples from RFs, but their approach still requires subtree retraining and suffers from several limitations in both clarity and design. Different from others, our proposed DYNFRS excels in both sequential and batch unlearning settings and supports learning new samples after training.

## 3 BACKGROUND

Our proposed framework is designed for classification and regression tasks. Let $\mathcal{D} \subseteq \mathbb{R}^d \times \mathcal{Y}$ represent the underlying sample space. The goal is to find a hypothesis $h$ that captures certain properties of the unknown $\mathcal{D}$ based on observed samples. We denote each sample by $\langle \mathbf{x}, y \rangle$, where $\mathbf{x} \in \mathbb{R}^d$ is a $d$-dimensional vector describing features of the sample and $y \in \mathcal{Y}$ represents the corresponding label or value. Denote $S$ as the set of observed samples, consisting of $n$ independent and identically distributed (i.i.d.) samples $\langle \mathbf{x}_i, y_i \rangle$ for $i \in [n]$ drawn from $\mathcal{D}$, where $[n] \triangleq \{i \in \mathbb{Z} \mid 1 \leq i \leq n\}$. For clarity, we call the $k$-th entry of $\mathbf{x}$ attribute $k$.

### 3.1 EXACT MACHINE UNLEARNING

The objective of machine unlearning for a specific learning algorithm $A$ is to efficiently forget certain samples. An additional constraint is that the unlearning algorithm must be equivalent to applying $A$ to the original dataset excluding the sample to be removed.

Formally, let the algorithm $A : \mathcal{S} \to \mathcal{H}$ maps a training set $S \in \mathcal{S}$ to a hypothesis $A(S) \in \mathcal{H}$. We then define $A^- : \mathcal{H} \times \mathcal{S} \times \mathcal{D} \to \mathcal{H}$ as an unlearning algorithm, where $A^-(A(S), S, \langle \mathbf{x}, y \rangle)$ produces the modified hypothesis with the impact of $\langle \mathbf{x}, y \rangle$ removed. The algorithm $A^-$ is termed an *exact* unlearning algorithm if the hypotheses $A(S \backslash \{\langle \mathbf{x}, y \rangle\})$ and $A^-(A(S), S, \langle \mathbf{x}, y \rangle)$ follow the same distribution. That is, for arbitrary hypothesis $h \in \mathcal{H}$, we have the following:

$$\Pr\left[A(S \backslash \{\langle \mathbf{x}, y \rangle\}) = h\right] = \Pr\left[A^-(A(S), S, \langle \mathbf{x}, y \rangle) = h\right]. \tag{1}$$

### 3.2 RANDOM FOREST

Prior to discussing Random Forests, it is essential to first introduce its base learner, the *Decision Tree (DT)*, a well-known tree-structured supervised learning model. It is proven that finding the optimal DT is NP-Hard (Hyafil and Rivest, 1976; Demirović et al., 2022); thus, studying hierarchical approaches is prevalent in the literature. In essence, the tree originates from a root containing all training samples and grows recursively by splitting a leaf node into new leaves until a predefined stopping criterion is met. Typically, DTs take the form of binary trees where each node branches into two by splitting $S_u$ (the set of samples obsessed by the node $u$) into two disjoint sets. Let $u_l$ and $u_r$ be the left and right child of node $u$, and we say split $(a, w)$ partitions $S_u$ into $S_{u_l}$ and $S_{u_r}$ if

$$S_{u_l} = \{\langle \mathbf{x}, y \rangle \in S_u \mid \mathbf{x}_a \leq w\}, \qquad S_{u_r} = \{\langle \mathbf{x}, y \rangle \in S_u \mid \mathbf{x}_a > w\}.$$

The best split $(a_u^\star, w_u^\star) \triangleq \arg\min\{I(S_u, (a, w)) \mid a \in [p], w \in \mathbb{R}\}$ is found among all possible splits by optimizing an empirical criterion score $I(S, \langle \mathbf{x}, y \rangle)$ such as the Gini index $I_G$ (Breiman et al., 1984), or the Shannon entropy $I_E$ (Quinlan, 1993):

$$I_G(S_u, (a, w)) = \sum_{v \in \{u_l, u_r\}} \frac{|S_v|}{|S_u|} \left(1 - \sum_{c \in \mathcal{Y}} \frac{|S_{v,c}|^2}{|S_v|^2}\right),$$

$$I_E(S_u, (a, w)) = \sum_{v \in \{u_l, u_r\}, c \in \mathcal{Y}} \left(-\frac{|S_{v,c}|}{|S_v|} \log_2 \frac{|S_{v,c}|}{|S_v|}\right),$$

where $S_{u,c} \triangleq \{\langle \mathbf{x}, y \rangle \in S_u \mid y = c\}$. The prediction for sample $\langle \mathbf{x}, y \rangle$ starts with the root and recursively goes down to a child until a leaf is reached, and the traversal proceeds to the left child if $\mathbf{x}_{a_u^\star} \leq w_u^\star$ and to the right otherwise.

*Random Forest (RF)* is an ensemble of independent DTs, where each tree is constructed with an element of randomness to enhance predictive performance. This randomness reduces the variance of the forest's predictions and thus lowers prediction error (Breiman, 2001). One method involves selecting the best split $(a_u^\star, w_u^\star)$ among $p$ randomly selected attributes rather than all $d$ attributes. Additionally, subsampling methods such as bootstrap (Breiman et al., 1984), or $m$-out-of-$n$ bootstrap (Genuer et al., 2017), are used to introduce more randomness. Bootstrap creates a training set $S^{(t)}$ for each tree $\varphi_t$ by drawing $n$ i.i.d. samples from the original dataset $S$ with replacement. A variant called $m$-out-of-$n$ bootstrap randomly picks $m$ different samples from $S$ to form $S^{(t)}$. These subsampling methods increase the diversity among trees, enhancing the robustness and generalizability of the model. However, all existing RF unlearning methods do not adopt subsampling, and Brophy and Lowd (2021) claims this exclusion does not affect the model's predictive accuracy.

### 3.3 EXTREMELY RANDOMIZED TREE

*Extremely Randomized Tree (ERT)* (Geurts et al., 2006) is a variant of the Decision Trees, but ERTs embrace more randomness when finding the best split. For a tree node $u$, both ERT and DT find the best splits on $p$ randomly selected attributes $a_{1\cdots p} \subseteq [d]$, but for each attribute $a_k$ ($k \in [p]$), ERT considers only $s$ candidates $\{(a_k, w_{k,i}) \mid i \in [s]\}$, where $w_{k,1\cdots s}$ are uniformly sampled from range $[\min\{\mathbf{x}_{i,a_k} \mid \langle \mathbf{x}_i, y_i \rangle \in S_u\}, \max\{\mathbf{x}_{i,a_k} \mid \langle \mathbf{x}_i, y_i \rangle \in S_u\}]$. Then, the best split $(a_u^\star, w_u^\star)$ is set as the candidates with optimal empirical criterion score (where $I$ could be either $I_G$ or $I_E$):

$$(a_u^\star, w_u^\star) \triangleq \arg\min\{I(S_u, (a_k, w_{k,i})) \mid k \in [p], i \in [s]\}.$$

Compared to DTs considering all $\mathcal{O}(p|S_u|)$ possible splits, ERTs consider only $\mathcal{O}(ps)$ candidates while maintaining similar predictive accuracy. This shrink in candidate size makes ERTs less sensitive to sample removal, and thus makes ERTs outstanding for efficient machine unlearning.

## 4 METHODS

In this section, we introduce the DYNFRS framework, which is structured into three components — the subsampling method OCC($q$), the lazy tag strategy LZY, and the base learner ERT. OCC($q$) allocates fewer training samples to each tree (i.e., $S^{(1)}, \cdots, S^{(T)}$) with the aim to minimize the work brought to each tree during both training and unlearning phrase while preserving par predictive accuracy. The lazy tag strategy LZY takes advantage of tree structure by caching and batching reconstruction needs within nodes and avoiding redundant work, and thus enables an efficient automatic tree structure modification and suiting DYNFRS for online fixed data streams. ERT require fewer adjustments towards sample addition/removal, making it the appropriate base learner for the framework. In a nutshell, DYNFRS optimize machine unlearning in tree-based methods from three perspectives — across trees (OCC($q$)), across requests (LZY), and within trees (ERT).

### 4.1 TRAINING SAMPLES SUBSAMPLING

Introducing divergence among DTs in the forest is crucial for enhancing predictive performance, as proven by Breiman (2001). Developing novel subsampling methods can further enhance this effect. Recall that $S$ represents the training set, and $S^{(t)}$ denotes the training set for the $t$-th tree $\varphi_t$. Empirical results (Section 5.2) indicate that having a reduced training set (i.e., $|S^{(t)}| < |S|$) does not degrade predictive performance and may even improve accuracy as more randomness is involved. In the following, we demonstrate that OCC($q$) leverages smaller $|S^{(t)}|$ and considerably shortens training and unlearning time.

The key idea is that if sample $\langle \mathbf{x}_i, y_i \rangle \in S$ does not occurs in tree $\varphi_t$ (i.e., $\langle \mathbf{x}_i, y_i \rangle \notin S^{(t)}$), then tree $\varphi_t$ is unaffected when unlearning $\langle \mathbf{x}_i, y_i \rangle$. Therefore, it is natural to constrain the occurrence of each sample $\langle \mathbf{x}_i, y_i \rangle \in S$ in the forest. To achieve this, OCC($q$) performs subsampling on trees instead of training samples. The algorithm starts with iterating through all training samples, and for each sample $\langle \mathbf{x}_i, y_i \rangle \in S$, $k$ different trees (say they are $\varphi_{i_1}, \ldots, \varphi_{i_k}$ with $i_{1\cdots k} \subseteq [n]$) are randomly and independently drawn from all $T$ trees (Algorithm 1: line 5), where $k$ is determined by the proportion factor $q$ ($q < 1$) satisfying $k = \lceil qT \rceil$. Then, OCC($q$) appends $\langle \mathbf{x}_i, y_i \rangle$ to all of the drawn tree's own training set ($S^{(i_1)}, S^{(i_2)}, \ldots, S^{(i_k)}$) (Algorithm 1: line 6-8). When all sample allocations finish, The

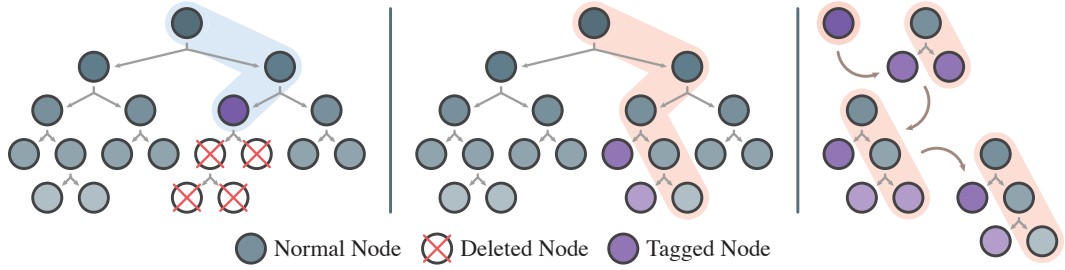

Normal Node    Deleted Node    Tagged Node

Figure 1: Left: (a) A sample addition/removal request arises. (b) Nodes it impacts are covered in the blue path. (c) There is a change in best split in the purple node (but not in other visited nodes), so a tag is placed on it. (d) The subtree of the tagged node is deleted. Middle: (a) A querying request arises. (b) Nodes that determine the prediction are covered in the orange path. (c) The tag is pushed down recursively until the query reaches a leaf. Right: A detailed process of how the querying request grows (split tagged node, push down its tag to its children recursively) the tree.

algorithm terminates in $\mathcal{O}(nk) = \mathcal{O}(nqT)$. Intuitively, $\text{OCC}(q)$ ensures that each sample occurs in exactly $k$ trees in the forest, and thus reduces the impact of each sample from all $T$ trees to merely $\lceil qT \rceil$ trees.

Further calculation confirms that $\text{OCC}(q)$ provides an expected $1/q^2$ unlearning speedup toward naïvely retraining. When unlearning an arbitrary sample with $\text{OCC}(q)$, without loss of generality (tree order does not matter as they are independent), assume it occurs in the first $k = \lceil qT \rceil$ trees $\varphi_1, \ldots, \varphi_k$. Calculation begins with finding the sum of all affected tree's training set sizes:

$$N_{\text{occ}} = \sum_{t=1}^{k} \left| S^{(t)} \right| = \sum_{i=1}^{|S|} \sum_{t=1}^{k} 1\left[ \langle \mathbf{x}_i, y_i \rangle \in S^{(t)} \right].$$

As for each sample $\langle \mathbf{x}_i, y_i \rangle \in S$, $\text{OCC}(q)$ assures that $\Pr[\langle \mathbf{x}_i, y_i \rangle \in S^{(t)}] = k/T = q$, giving,

$$\mathbb{E}\left[N_{\text{occ}}\right] = \sum_{i=1}^{|S|} \sum_{t=1}^{k} \mathbb{E}\left[ 1\left[ \langle \mathbf{x}_i, y_i \rangle \in S^{(t)} \right] \right] = \sum_{i=1}^{|S|} \sum_{t=1}^{k} \Pr\left[ \langle \mathbf{x}_i, y_i \rangle \in S^{(t)} \right] = qk|S| = q^2 T|S|.$$

For the naïve retraining method, the corresponding sum of affected sample size is $T|S|$ ($N_{\text{nai}} = T|S|$) as all trees are affected. As the retraining time complexity is linear to $N_{\text{occ}}$ and $N_{\text{nai}}$ (Appendix A.2, Theorem 2), the expected computational speedup provided by $\text{OCC}(q)$ is $\mathbb{E}[N_{\text{nai}}/N_{\text{occ}}] = 1/q^2$, when $\text{OCC}(q)$ and the naïve method adopt the same retraining method.

The above analysis also concludes that training a Random Forest with $\text{OCC}(q)$ will result in a $1/q$ times boost. In practice, we will empirically show (in Section 5.2) that by taking $q = 0.1$ or $q = 0.2$, the resulting model would have a similar or even higher accuracy in binary classification, which benefits unlearning with a $100\times$ or $25\times$ boost, and make training $10\times$ or $5\times$ faster.

## 4.2 LAZY TAG STRATEGY

There are two observations on tree-based methods that make LZY possible. (1) During the unlearning phase, the portion of the model that is affected by the deleted sample is known (Fig. 1 left, blue path, and deleted nodes). (2) During the inference phase, only a small portion (a tree path) of the whole model determines the prediction of the model (Fig. 1 middle, orange path) (Louppe, 2015). Along with another universal observation (3) Adjustments to the model is unnecessary until a querying request arises, we are able to develop the LZY lazy tag strategy that minimizes adjustments to tree structure when facing a mixture of sample addition/removal and querying requests. In contrast, neural network based methods do not fully possess the properties mentioned in observation (1) and (2), although encouraging exploration on similar properties has been reported (Morcos et al., 2018; Goh et al., 2021; Zhu et al., 2024).

Intuitively, A tree node $u$ can remain unchanged if adding/removing a sample does not change its best split $(a_u^\star, w_u^\star)$. Then, the sample addition/removal request would affect only one of the $u$'s

children since the best split partitions $u$ into two disjoint parts. Following this process recursively, the request will go deeper in the tree until a leaf is reached or a change in best split occurs. Unfortunately, if the change in best split occurs in node $u$, we need to retrain the whole subtree rooted by $u$ to meet the criterion for exact unlearning. Therefore, a sample addition/removal request affects only a path and a subtree of the whole tree, which is observation (1).

As retraining a subtree is time-consuming, we place a tag on $u$, denoting it needs a reconstruction (Algorithm 2: line: 8-9). When another sample addition/removal request reaches the tagged $u$ later, it just simply ends there (Algorithm 2: line 7) since the will-be-happening reconstruction would cover this request as retraining is the most effective unlearning method. But when a querying request meets a tag, we need to lead it to a leaf for an accurate prediction (observation (3)). As we do not retrain the subtree, and recall that observation (2) states that the query only observes a path in the tree, so the minimum effect to fulfill this query is to reconstruct the tree path that connects $u$ and a leaf, instead of the entire subtree $u$. To make this happen, we find the best split of $u$ and grow the left and right child. Then, we clear the tag on $u$ since it has been split and push down the tags to both of its children, indicating further splitting on them is needed (Algorithm 2: line 26-27). As depicted in Fig. 1 right side, the desired path reveals when the recursive pushing-down process reaches a leaf.

To summarize, only queries activate node reconstruction within a node, and LZY automatically batches all node reconstructions between two visiting queries, saving plenty of computational efforts. From the tree's perspective, LZY replaces the subtree retraining by amortizing it into path constructions in queries so that the latency for responding to modification requests is reduced. Unlike OCC($q$), which takes advantage of ensemble and brings less workload to each tree, LZY relies on tree structures, dismantling requests into smaller parts and handling these parts in batches for smaller workload.

## 4.3 UNLEARNING IN EXTREMELY RANDOMIZED TREES

Despite OCC($q$) and LZY making no assumption on the forest's base learner, we opt for Extremely Randomized Trees (ERTs) with the aim of achieving the best performance in machine unlearning. Different from Decision Trees, ERTs are more robust to changes in training samples while remaining competitive in predictive performance. This robustness ensures the whole DYNFRS framework undergoes fewer changes in tree structures when unlearning.

In essence, each ERT node finds the best split among $s$ (usually around 5 to 20) candidates on one attribute instead of all possible splits (possibly more than $10^5$) so that the best split has a higher chance to remain unchanged when a sample addition/removal occurs in that node. Additionally, It takes a time complexity of $\mathcal{O}(ps)$ for ERTs to detect whether the change in best split occurs if all candidate's split statistics are stored during the training phase, which is much more efficient than the $\mathcal{O}(p|S_u|)$ detection for Decision Trees. To be specific, for each ERT node $u$, we store a subset of attributes $a_{1\cdots p} \subseteq [d]$, and for each interested attribute $a_k$ ($k \in [p]$), $s$ different thresholds $w_{k,1\cdots s}$ are randomly generated and stored. Therefore, a total of $p \cdot s$ candidates $\{(a_k, w_{k,i}) \mid k \in [p], i \in [s]\}$ determine the best split of the node. Furthermore, the split statistics of each candidate $(a_k, w_{k,i})$ are also kept, which consists of its empirical criterion score, the number of samples less than the threshold, and the number of positive samples less than the threshold. When a sample addition/removal occurs, each candidate's split statistics can be updated in $\mathcal{O}(1)$, and we assign the one with optimal empirical criterion score as the node's best split.

However, one special case is that when a change in range of $a_k$ occurs, a resampling on $w_{k,1\cdots s}$ is needed. ERT node $u$ find the candidates of attribute $a_k$ by generating i.i.d. samples following a uniform distribution on $[a_{k,\min}, a_{k,\max}]$, where $a_{k,\min} \triangleq \min\{\mathbf{x}_{i,a_k} \mid \langle \mathbf{x}_i, y_i \rangle \in S_u\}$ and $a_{k,\max}$ is defined similarly. Therefore, we keep track of $a_{k,\min}$ and $a_{k,\max}$ and resample candidates' threshold when the range $[a_{k,\min}, a_{k,\max}]$ changes due to sample addition/removal.

## 4.4 THEORETICAL RESULTS

Due to the page limit, all detailed proofs of the following theorems are provided in Appendix A.2.

We first demonstrate that DYNFRS's approach to sample addition and removal suits the definition of exact (un)learning (Section 3.1), validating the unlearning efficacy of DYNFRS:

**Theorem 1.** *Sample addition and removal for the* DYNFRS *framework are exact.*

Next, we establish the theoretical bound for time efficiency of DYNFRS across different aspects. Conventionally, for an ERT node $u$ and a certain attribute $a$, finding the best split of attribute $a$ requires a time complexity of $\mathcal{O}(|S_u| \log |S_u|)$. In this work, we propose a more efficient $\mathcal{O}(|S_u| \log s)$ algorithm (see Appendix A.2, Lemma 1) to find the desired split by turning it into a range addition and query problem. Since $|S_u| \gg s$ in most cases, and usually $\log_2 s \leq 5$, this new algorithm is advantageous in both theoretical bounds and practical performance.

For DYNFRS with $T$ trees, each having a maximum depth $d_{\max}$, and considering $p$ attributes per node, the time complexity for training on a training set with $n = |S|$ is derived as:

**Theorem 2.** *Training* DYNFRS *yields a time complexity of* $\mathcal{O}(qTd_{\max}pn \log s)$.

DYNFRS also achieves an outstanding time complexity for sample addition/removal. For clarity, we define $n_{\text{aff}}$ as the sum of sample size ($|S_u|$) of all node $u$ s.t. $u$ is met by the sample addition/removal request, and a change in range occurs, and $c$ be the number of attributes whose range has changed. Further, denote $n_{\text{lzy}}$ as the sum of sample size ($|S_u|$) of all node $u$ s.t. $u$ is tagged due to this request. Based on observations described in Section 4.2, we claim the following:

**Theorem 3.** *Modification (sample addition or removal) in* DYNFRS *yields a time complexity of* $\mathcal{O}(qTd_{\max}ps)$ *if no attribute range changes occur while* $\mathcal{O}(qTd_{\max}ps + cn_{aff} \log s)$ *otherwise.*

**Theorem 4.** *Query in* DYNFRS *yields a time complexity of* $\mathcal{O}(Td_{\max})$ *if no lazy tag is met, while* $\mathcal{O}(Td_{\max} + pn_{lzy} \log s)$ *otherwise.*

## 5 EXPERIMENTS

In this section, we empirically evaluate the DYNFRS framework on the predictive performance, machine unlearning efficiency, and response latency in the online mixed data stream setting.

### 5.1 IMPLEMENTATION

Due to the page limit, this part is moved to Appendix A.3.

#### 5.1.1 BASELINES

We use DaRE (Brophy and Lowd, 2021), HedgeCut (Schelter et al., 2021), and OnlineBoosting (Lin et al., 2023) as baseline models. Although OnlineBoosting employs a different learning algorithm, it is included due to its superior performance in batch unlearning. Additionally, we included the Random Forest Classifier implementation (we call it *Vanilla* in tables below) from scikit-learn to provide an additional comparison of predictive performance. For all baseline models, we adhere to the instructions provided in the original papers and use the same parameter settings. More details regarding the baselines are in Appendix A.4.

#### 5.1.2 DATASETS

Table 1: Datasets specifications. (# train: number of training samples; # test: number of testing samples; % pos: percentage of positive samples; # attr: number of attributes; # attr-hot: number of attributes after one-hot encoding; # cat: number of categorical attributes.)

| Datasets | # train | # test | % pos | # attr | # attr-hot | # cat |
|---|---|---|---|---|---|---|
| Purchase | 9864 | 2466 | .154 | 17 | 17 | 0 |
| Vaccine | 21365 | 5342 | .466 | 36 | 184 | 36 |
| Adult | 32561 | 16281 | .239 | 13 | 107 | 8 |
| Bank | 32950 | 8238 | .113 | 20 | 63 | 10 |
| Heart | 56000 | 14000 | .500 | 12 | 12 | 0 |
| Diabetes | 81412 | 20354 | .461 | 43 | 253 | 36 |
| NoShow | 88421 | 22106 | .202 | 17 | 98 | 2 |
| Synthetic | 800000 | 200000 | .500 | 40 | 40 | 0 |
| Higgs | 8800000 | 2200000 | .530 | 28 | 28 | 0 |

We test DYNFRS on 9 binary classification datasets that vary in size, positive sample proportion, and attribute types. The technical details of these data sets can be found in Table 1. For better predictive performance, we apply one-hot encoding for categorical attributes (attributes whose values are discrete classes, such as country, occupation, etc.). Further details regarding datasets are offered in the Appendix A.5.

### 5.1.3 METRICS

For predictive performance, we evaluate all the models with accuracy (number of correct predictions divided by the number of tests) if the dataset has less than $21\%$ positive samples or AUC-ROC (the area under receiver operating characteristic curve) otherwise.

To evaluate models' unlearning efficiency, we follow Brophy and Lowd (2021) and use the term *boost* standing for the speedup relative to the naïve retraining approach (i.e., the number of samples unlearned when naïvely retraining unlearns 1 sample). Each model's naïve retraining procedure is implemented in the same programming language as the model itself. Additionally, we report the time elapsed during the unlearning process for direct comparison.

### 5.2 PREDICTIVE PERFORMANCE

Table 2: Predictive performance (↑) comparison between models. Each cell contains the accuracy or AUC-ROC score with standard deviation in a smaller font. The best-performing model is bolded.

| Datasets | DaRE | HedgeCut | Vanilla | Online Boosting | DYNFRS ($q = 0.1$) | DYNFRS ($q = 0.2$) |
|---|---|---|---|---|---|---|
| Purchase | .9327±.001 | .9118±.001 | **.9372**±.001 | .9207±.000 | .9327±.001 | .9359±.001 |
| Vaccine | .7916±.003 | .7706±.002 | .7939±.002 | **.8012**±.000 | .7911±.001 | .7934±.002 |
| Adult | .8628±.001 | .8428±.001 | .8637±.000 | .8503±.000 | .8633±.001 | **.8650**±.001 |
| Bank | .9420±.000 | .9350±.000 | .9414±.001 | **.9436**±.000 | .9417±.001 | **.9436**±.000 |
| Heart | .7344±.001 | .7195±.001 | .7342±.001 | .7301±.000 | .7358±.002 | **.7366**±.000 |
| Diabetes | .6443±.001 | .6190±.000 | .6435±.001 | .6462±.000 | .6453±.001 | **.6470**±.002 |
| NoShow | .7361±.001 | .7170±.000 | **.7387**±.000 | .7269±.000 | .7335±.001 | .7356±.000 |
| Synthetic | .9451±.000 | / | .9441±.000 | .9309±.000 | .9424±.000 | **.9454**±.000 |
| Higgs | .7441±.000 | / | .7434±.000 | .7255±.000 | .7431±.000 | **.7475**±.000 |

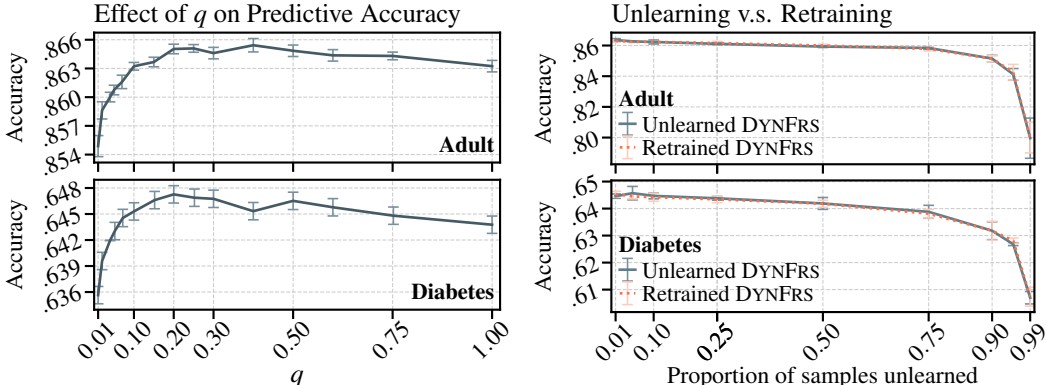

Figure 2: DYNFRS with different $q$ is tested on the dataset Adult and Diabetes, and the tendency is shown by the curve with the standard deviation shown by error bars.

Figure 3: DYNFRS's unlearning exactness. The blue-gray solid line represents the accuracy tendency of the unlearned model, while the orange dotted line represents that of the retrained model.

We evaluate the predictive performance of DYNFRS in comparison with 4 other models in 9 datasets. The detailed results are listed in Table 2. In 6 out of 9 datasets, DYNFRS($q = 0.2$) outperforms all other models in terms of predictive performance, while OnlineBoosting shows an advantage in the Vaccine and Bank dataset, and scikit-learn Random Forest comes first in Purchase and NoShow.

These results show that $\text{OCC}(0.2)$ sometimes improves the forest's predictive performance. Looking closely at $\text{DYNFRS}(q = 0.1)$, we observe that its accuracy is similar to that of DaRE and the scikit-learn Random Forest. All the hyperparameters used in $\text{DYNFRS}$ are listed in the Appendix A.12.

The effects of $q$ are assessed on the two most commonly tested datasets (Adult and Diabetes) in RF classification. From Fig. 2, an acute drop in predictive accuracy is obvious when $q < 0.1$. For the Adult dataset, $\text{DYNFRS}$'s predictive accuracy peaks at $q = 0.4$, while a similar tendency is observed for the Diabetes dataset with the peak at $q = 0.2$. However, to avoid tuning on $q$, we suggest choosing $q = 0.2$ to optimize accuracy and $q = 0.1$ to improve unlearning efficiency.

To determine whether $\text{DYNFRS}(q = 0.1)$ achieves exact unlearning (Section 3.1), we compare it with a retrain-from-scratch model with different sample removal proportions. Specifically, Let $S^-$ denote the set of samples requested for removal, and we compare the predictive performance of an unlearned model — trained on complete training set $S$ and subsequently unlearning all samples in $S^-$ — with the retrained model, which is trained directly on $S \backslash S^-$. Note that both models adopt the same training algorithm. As depicted in Fig. 3, the performance of both models is nearly identical across different removal proportions (i.e., $|S^-|/|S|$). This close alignment suggests that, empirically, the unlearned model and the retrained model follow the same distribution (Equation 1).

## 5.3 SEQUENTIAL UNLEARNING

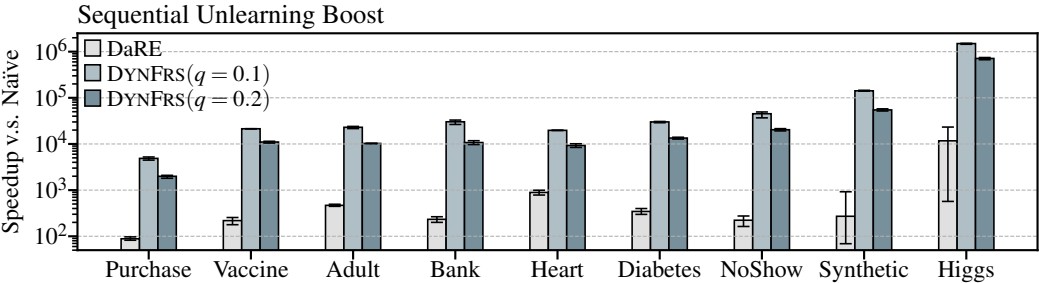

Figure 4: Comparison of sequential unlearning boost ($\uparrow$) between $\text{DYNFRS}$ with different $q$ and DaRE, and error bars represent the minimum and maximum values among five trials.

In this section, we evaluate the efficiency of $\text{DYNFRS}$ in sequential unlearning settings, where models process unlearning requests individually. In experiments, models are fed with a sequence of unlearning requests until the time elapsed exceeds the time naïve retraining model unlearns one sample. To ensure that $\text{DYNFRS}$ does not gain an unfair advantage by merely tagging nodes without modifying tree structures, we disable $\text{LZY}$ and use a Random Forest without $\text{OCC}(q)$ as the naïve retraining method for $\text{DYNFRS}$. As shown in Fig. 4, $\text{DYNFRS}$ consistently outperforms DaRE, the state-of-the-art method in sequential unlearning for RFs, across all datasets in both $q = 0.1$ and $q = 0.2$ settings, and achieves a $22\times$ to $523\times$ speedup relative to DaRE. Furthermore, $\text{DYNFRS}$ demonstrates a more stable performance compared to DaRE who exhibits large error bars in Higgs.

HedgeCut is excluded from the plot as it is unable to unlearn more than $0.1\%$ of the training set, making boost calculation often impossible. OnlineBoosting is also omitted due to poor performance, achieving boosts of less than 10. This inefficiency stems from its slow unlearning efficiency of individual instances (see Fig. 5 upper-left plot). Appendix A.6 contains more experiment results.

## 5.4 BATCH UNLEARNING

We measure each model's batch unlearning performance based on execution time ($\text{DYNFRS}$'s runtime includes retraining all the tagged subtree), where one request contains multiple samples to be unlearned. The results indicate that $\text{DYNFRS}$ significantly outperforms other models across all datasets and batch sizes (see Appendix A.6 for complete results). In the lower-left and lower-right plots of Fig. 5, DaRE demonstrates excessive time requirements for unlearning $0.1\%$ and $1\%$ of samples in the large-scale datasets Synthetic and Higgs, primarily due to its inefficiency when dealing large batches. In contrast, OnlineBoosting achieves competitive performance on these datasets, but $\text{DYNFRS}$ completes the same requests in half the time. Furthermore, OnlineBoosting shows

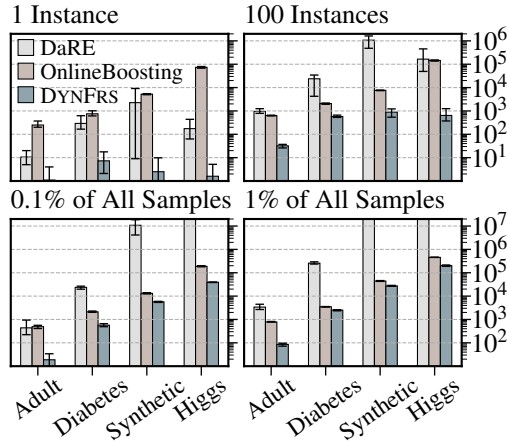
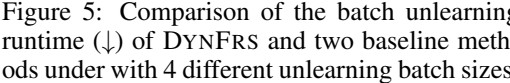
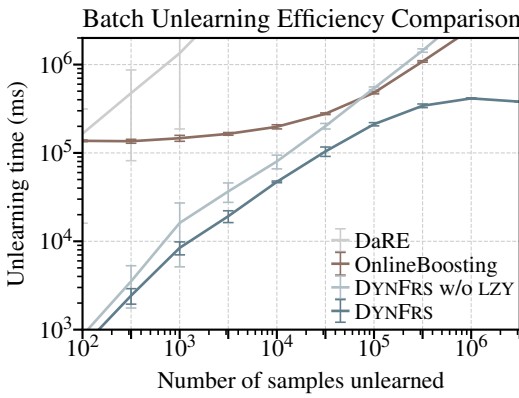

Figure 5: Comparison of the batch unlearning runtime (↓) of DYNFRS and two baseline methods under with 4 different unlearning batch sizes.

Figure 6: Comparison of the batch unlearning runtime tendency (↓) between models using different numbers of unlearned samples in Higgs. Runtimes are measured in milliseconds (ms).

the poorest performance in single-instance unlearning (Fig. 5 upper-left), and this inability limits its effectiveness in sequential unlearning. Overall, Fig. 5 demonstrates that DYNFRS is the only model that excels in both sequential and batch unlearning contexts.

Further investigation of each model's behavior across different unlearning batch sizes is presented in Fig. 6. Both DaRE and DYNFRS w/o LZY exhibit linear trends in the plot, indicating their lack of specialization for batch unlearning. Meanwhile, the curve of OnlineBoosting maintains a stationary performance for batch sizes up to $10^4$ but experiences a rapid increase in runtime beyond this threshold. Notably, the curve for DYNFRS stays below those of all other models, demonstrating its advantage across all batch sizes in dataset Higgs. Additionally, DYNFRS is the only model whose runtime converges for large batches, attributed to the presence of LZY.

## 5.5 ONLINE MIXED DATA STREAM

In this section, we introduce the online mixed data stream setting, which satisfies: (1) there are 3 types of requests: sample addition, sample removal, and querying; (2) requests arrive in a mixed sequence, with no prior knowledge of future requests until the current one is processed; (3) the amount of addition and removal requests are roughly balanced, allowing unchanged model's hyperparameters; (4) the goal is to minimize the latency in responding to each request. Currently, no other tree-based models but DYNFRS can handle sample addition/removal and query simultaneously.

Due to the page limit, this part is moved to Appendix A.10.

## 6 CONCLUSION

In this work, we introduced DYNFRS, a framework that supports efficient machine unlearning in Random Forests. Our results show that DYNFRS is 4-6 orders of magnitude faster than the naïve retraining model and 2-3 orders of magnitude faster than the state-of-the-art Random Forest unlearning approach DaRE (Brophy and Lowd, 2021). DYNFRS also outperforms OnlineBoosting (Lin et al., 2023) in batch unlearning. In the context of online data streams, DYNFRS demonstrated strong performance, with an average latency of 0.12 ms on sample addition/removal in the large-scale dataset Higgs. This efficiency is due to the combined effects of the subsampling method OCC($q$), the lazy tag strategy LZY, and the robustness of Extremely Randomized Trees, bringing Random Forests closer to real-world applicability in dynamic data environments.

For future works, we will investigate more strategies that take advantage of the tree structure and accelerate Random Forest in the greatest extent for real-world application.

## 7 REPRODUCIBILITY STATEMENT

- **Code**: We provide pseudocode to help understand this work in Appendix A.1. All our code is publicly available at: https://github.com/shurongwang/DynFrs.

- **Datasets**: All datasets are either included in the repo, or a description for how to download and preprocess the dataset is provided. All datasets are public and raise no ethical concerns.

- **Hyperparameters**: All parameters of our proposed framework are provided in Appendix A.12, Table 15.

- **Environment**: Details of our experimental setups are provided in Appendix A.3.

- **Random Seed**: we use C++'s mt19937 module with a random device for all random behavior, with the random seed determined by system time.

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

# A APPENDIX

## A.1 PSEUDOCODE

**Algorithm 1** Training DYNFRS and Unlearning Samples with OCC($q$)

1: **procedure** DISTRIBUTE($S, T, q$)
2:     $k \leftarrow T \times q$
3:     $S^{(1)}, \ldots, S^{(T)} \leftarrow \varnothing, \cdots, \varnothing$
4:     **for** $\langle \mathbf{x}_i, y_i \rangle \in S$ **do**
5:         $j_{1 \ldots k} \leftarrow$ randomly and independently sample $k$ different integers from $[T]$
6:         **for** $t \in j_{1 \ldots k}$ **do**
7:             $S^{(t)} \leftarrow S^{(t)} \cup \{\langle \mathbf{x}_i, y_i \rangle\}$
8:         **end for**
9:     **end for**
10:     **return** $S^{(1)}, \ldots, S^{(T)}$
11: **end procedure**
12:
13: **procedure** TRAIN($S, T, q$)
14:     $\Phi \leftarrow \varnothing$
15:     $S^{(1)}, \ldots, S^{(T)} \leftarrow$ DISTRIBUTE($S, T, q$)
16:     **for** $t \leftarrow 1 \cdots T$ **do**
17:         $\varphi_t \leftarrow$ BUILDTREE($S^{(t)}$)
18:         $\Phi \leftarrow \Phi \cup \{\varphi_t\}$
19:     **end for**
20:     **return** $\Phi$
21: **end procedure**
22:
23: **procedure** ADD($\Phi, \langle \mathbf{x}, y \rangle$)
24:     $j_{1 \ldots k} \leftarrow$ randomly and independently sample $k$ different integers from $[T]$
25:     **for** $t \leftarrow j_{1 \ldots k}$ **do**
26:         ADD($\varphi_t, \langle \mathbf{x}, y \rangle$)
27:     **end for**
28: **end procedure**
29:
30: **procedure** REMOVE($\Phi, \langle \mathbf{x}, y \rangle$)
31:     **for** $t \leftarrow 1 \cdots T$ **do**
32:         **if** $\langle \mathbf{x}, y \rangle \in S^{(t)}$ **then**
33:             REMOVE($\varphi_t, \langle \mathbf{x}, y \rangle$)
34:         **end if**
35:     **end for**
36: **end procedure**

**Algorithm 2** Unlearning and Querying in Trees with LZY

1: **procedure** REMOVE($\varphi, \langle \mathbf{x}, y \rangle$)   ▷ ADD is similar
2:     DELETE(ROOT($\varphi$), $\langle \mathbf{x}, y \rangle$)
3: **end procedure**
4:
5: **procedure** REMOVE($u, \langle \mathbf{x}, y \rangle$)
6:     $S_u \leftarrow S_u \backslash \langle \mathbf{x}, y \rangle$   ▷ implementation of DYNFRS does not actually store $S_u$, so this line stands for updating all split statistics of node $u$
7:     **if** LZY$_u = 1$ **then return**
8:     **else if** BESTSPLITCHANGED($u$) **then**
9:         LZY$_u \leftarrow 1$
10:     **else if** $\neg$ISLEAF($u$) **then**
11:         **if** $\mathbf{x}_{a_u^\star} \leq w_u^\star$ **then**
12:             REMOVE($u_l, \langle \mathbf{x}, y \rangle$)
13:         **else**
14:             REMOVE($u_r, \langle \mathbf{x}, y \rangle$)
15:         **end if**
16:     **end if**
17: **end procedure**
18:
19: **procedure** QUERY($\varphi, \langle \mathbf{x}, y \rangle$)
20:     QUERY(ROOT($\varphi$), $\langle \mathbf{x}, y \rangle$)
21: **end procedure**
22:
23: **procedure** QUERY($u, \langle \mathbf{x}, y \rangle$)
24:     **if** LZY$_u = 1$ **then**
25:         SPLIT($u$)
26:         LZY$_u \leftarrow 0$
27:         LZY$_{u_l} \leftarrow 1$, LZY$_{u_r} \leftarrow 1$
28:     **end if**
29:     **if** ISLEAF($u$) **then**
30:         **return** $S_u$
31:     **end if**
32:     **if** $\mathbf{x}_{a_u^\star} \leq w_u^\star$ **then**
33:         QUERY($u_l, \langle \mathbf{x}, y \rangle$)
34:     **else**
35:         QUERY($u_r, \langle \mathbf{x}, y \rangle$)
36:     **end if**
37: **end procedure**

## A.2 PROOFS

**Theorem 1.** *Sample addition and removal for the* DYNFRS *framework are exact.*

*Proof.* We prove that the subsampling method OCC($q$) maintains the exactness of DYNFRS. Let random variable $o_{i,t} \triangleq [\langle \mathbf{x}_i, y_i \rangle \in S^{(t)}]$ denotes whether $\langle \mathbf{x}_i, y_i \rangle$ occurs in $S^{(t)}$. In OCC($q$), each sample $\langle \mathbf{x}_i, y_i \rangle$ is distributed to $\lceil qT \rceil$ distinct trees, with the selection of these trees being independent of other samples $\langle \mathbf{x}_j, y_j \rangle \in S$ for $j \neq i$. Thus, $o_{i,\cdot}$ is independent from $o_{j,\cdot}$ for $j \neq i$. However, $o_{i,1}, o_{i,2}, \ldots, o_{i,T}$ are dependent on each other, constrained by $\forall t \in [T], o_{i,t} \in \{0, 1\}$, $\sum_{t=0}^{T} o_{i,t} = \lceil qT \rceil$, and we say they follow a joint distribution $\mathcal{B}(T, q)$.

Now, let $S'^{(1)}, S'^{(2)}, \ldots, S'^{(T)}$ denotes the training sets for each tree generated by applying OCC($q$) to the modified training set $S'$, and let $o'_{i,t} \triangleq [\langle \mathbf{x}_i, y_i \rangle \in S'^{(t)}]$. Then, when removing sample

$\langle \mathbf{x}_i, y_i \rangle$ (i.e., $S' = S \backslash \langle \mathbf{x}_i, y_i \rangle$), we have $(o_{j,1}, \ldots, o_{j,T}) \sim \mathcal{B}(T, q)$ and $(o'_{j,1}, \ldots, o'_{j,T}) \sim \mathcal{B}(T, q)$ for $i \neq j$, and $o_{i,\cdot} = 0$ (because $\langle \mathbf{x}_i, y_i \rangle$ is removed) and $o'_{i,\cdot} = 0$. Notably, $\mathcal{B}(T, q)$ depends on $T$ and $q$ only, but not on training samples $S$. This shows that simply setting $o_{i,\cdot} = 0$ ensures that $\text{OCC}(q)$ with sample removed maintains the same distribution as applying $\text{OCC}(q)$ on the modified training set.

Similarily, when adding $\langle \mathbf{x}_i, y_i \rangle$ (i.e., $S' = S \cup \langle \mathbf{x}_i, y_i \rangle$), $(o_{j,1}, \ldots, o_{j,T})$ and $(o'_{j,1}, \ldots, o'_{j,T})$ follow the same distribution $\mathcal{B}(T, q)$ for $j \neq i$. While $(o_{j,1}, \ldots, o_{j,T})$ is generated from $\mathcal{B}(T, q)$ for addition, it is equivalent to $(o'_{i,1}, \ldots, o'_{i,T})$ for following the same distribution. Therefore, $\text{OCC}(q)$ with sample added maintains the same distribution as applying $\text{OCC}(q)$ on the modified training set.

Next, we prove that the addition and removal operations are exact within a specific DYNFRS tree. When no change in range of attribute $a$ ($[\min\{\mathbf{x}_{i,a} \mid \langle \mathbf{x}_i, y_i \rangle \in S_u\}, \max\{\mathbf{x}_{i,a} \mid \langle \mathbf{x}_i, y_i \rangle \in S_u\}]$) occurs, the candidate splits are sampled from the same uniform distribution making DYNFRS and the retraining method are identical in distribution node's split candidates. However, when a change in range occurs, DYNFRS resamples all candidate splits and makes them stay in the same uniform distribution as those in the retraining method. Consequently, DYNFRS adjusts itself to remain in the same distribution with the retraining method. Thus, sample addition and removal in DYNFRS are exact.

$\square$

**Lemma 1.** *For a certain Extremely Randomized Tree node $u$, and a specific attribute $a$, the time complexity of finding the best split of attribute $a$ is $\mathcal{O}(|S_u| \log s)$, assuming that $|S_u| \gg s$.*

*Proof.* Conventionally, for each tree node $u$ and an attribute $a$, we uniformly samples $s$ thresholds $w_{a,1 \ldots s}$ from $[\min\{\mathbf{x}_{i,a} \mid \langle \mathbf{x}_i, y_i \rangle \in S_u\}, \max\{\mathbf{x}_{i,a} \mid \langle \mathbf{x}_i, y_i \rangle \in S_u\}]$. Then, we try to split $S_u$ with each threshold and look for split statistics that are: (1) the number of samples in the left or right child (i.e., $|S_{u_l}|$ and $|S_{u_r}|$), and (2) the number of positive samples in left or right child ($|S_{u_l,+}|$ and $|S_{u_r,+}|$), which are the requirements for calculating the empirical criterion scores.

One approach, as used by prior works, first sort all samples $S_u$ by $\mathbf{x}_{\cdot,a}$ in ascending order, and then sort thresholds $w_{a,1 \ldots s}$ in ascending order. These sortings has a time complexity of $\mathcal{O}(|S_u| \log |S_u|)$ and $\mathcal{O}(s \log s)$, respectively. After that, a similar technique used in the merge sort algorithm is used to find the desired split statistics in $\mathcal{O}(|S_u| + s)$.

To get rid of the costly sorting on $S_u$, we sort $w_{a,1 \ldots s}$ $\mathcal{O}(s \log s)$ and then iterate through all samples and calculate the changes each sample brings to candidates' split statistics. For convenience, let

$$b_k \triangleq \big|\{\langle \mathbf{x}_i, y_i \rangle \mid \langle \mathbf{x}_i, y_i \rangle \in S_u \wedge \mathbf{x}_{i,a} \leq w_{a,k}\}\big|,$$
$$c_k \triangleq \big|\{\langle \mathbf{x}_i, y_i \rangle \mid \langle \mathbf{x}_i, y_i \rangle \in S_u \wedge \mathbf{x}_{i,a} \leq w_{a,k} \wedge y_i = +\}\big|,$$

which are crucial split statistics for calculating the empirical criterion score. We start with setting $b_{1 \ldots s}$ and $c_{1 \ldots s}$ as all zeros. Then, for a sample $\langle \mathbf{x}_i, y_i \rangle \in S_u$, it will cause an increment in $b_{s' \ldots s}$ for some $s'$ satisfying $\mathbf{x}_{i,a} \leq w_{s'}$ and $\mathbf{x}_{i,a} > w_{s'-1}$. Given that $w_{a,1 \ldots s}$ are sorted, all $k$, $(s' \leq k \leq s)$ satisfy $\mathbf{x}_{i,a} \leq w_{a,k}$, while $\mathbf{x}_{i,a} > w_{a,k}$ for all $1 \leq k < s'$. $s'$ can be easily found by binary search in $\mathcal{O}(\log s)$, then adding 1 to $b_{s'}, b_{s'+1}, \ldots, b_s$ is the only thing left. Use a loop for range addition is clearly $\mathcal{O}(s)$, but insteading of finding $b_{1 \ldots s}$, we keep track of $d_{1 \ldots s}$, where $d_k \triangleq b_k - b_{k-1}$. So increment $b_{s' \ldots s}$ can be replace by $d_{s'} \leftarrow d_{s'} + 1$, which is $\mathcal{O}(1)$. When all samples are processed, we construct $b_{1 \cdot s}$ from $d_{1 \ldots s}$, where prefix sums $b_k \leftarrow b_{k-1} + d_k$ help solve it in $\mathcal{O}(s)$.

For every sample $\langle \mathbf{x}_i, y_i \rangle \in S$, we need to find $s'$ in $\mathcal{O}(\log s)$ (Algorithm 3: line 10), and perform increment in $d'_s$ in $\mathcal{O}(1)$ (Algorithm 3: line 11), which results in a time complexity of $\mathcal{O}(|S_u| \log s)$ in this part (Algorithm 3: line 9-14). Meanwhile, the prefix sum is executed after all samples are processed (Algorithm 3: line 15-18), and its execution time is bounded by $\mathcal{O}(s)$. Luckily, $c_{1 \ldots s}$ can be calculated in a similar manner, and with both $b_{1 \ldots s}$ and $c_{1 \ldots s}$ ready, we can obtain the empirical criterion score for each candidate split (Algorithm 3: line 21-28), and this has a time complexity of $\mathcal{O}(s)$ assuming calculating criterion scores to be $\mathcal{O}(1)$.

Since $|S_u| \gg s$, the term $\mathcal{O}(|S_u| \log s)$ dominates in time complexity, with the binary search (Algorithm 3: line 10) being the threshold. It is noteworthy that adopting exponential search to find $s'$ can result in an expected $\mathcal{O}(\log \log s)$ time complexity since $w_{a,1 \ldots s}$ are uniformly distributed.

However, binary search outperforms exponential search in practice, so we conclude with a time complexity of $\mathcal{O}(|S_u| \log s)$ for finding the best split of attribute $a$ in node $u$ when $|S_u| \gg s$.

---

**Algorithm 3** Find the best split for attribute $a$

1: **procedure** FINDATTRIBUTEBESTSPLIT$(S_u, a, s)$
2:     $R \leftarrow [\min\{\mathbf{x}_{i,a} \mid \langle \mathbf{x}_i, y_i \rangle \in S_u\}, \max\{\mathbf{x}_{i,a} \mid \langle \mathbf{x}_i, y_i \rangle \in S_u\}]$
3:     $w_{a,1\cdots s} \leftarrow$ sample $s$ i.i.d. values from $R$
4:     $w_{a,1\cdots s} \leftarrow$ SORT$(w_{a,1\cdots s})$                                       $\triangleright$ sort $w_{a,1\cdots s}$ in ascending order.
5:     $b_{1\cdots s} \leftarrow \{0, \cdots, 0\}$                                          $\triangleright$ create an 1-D array of size $s$
6:     $c_{1\cdots s} \leftarrow \{0, \cdots, 0\}$                                          $\triangleright$ create an 1-D array of size $s$
7:     $n \leftarrow |S_u|$
8:     $n_+ \leftarrow 0$
9:     **for** $\langle \mathbf{x}_i, y_i \rangle \in S_u$ **do**
10:         $s' \leftarrow$ LOWERBOUND$(w_{1\cdots s}, \mathbf{x}_{i,a})$               $\triangleright$ find the largest position $k$ s.t. $w_{k-1} < \mathbf{x}_{i,a}$
11:         $b_{s'} \leftarrow b_{s'} + 1$
12:         $c_{s'} \leftarrow c_{s'} + y_i$
13:         $n_+ \leftarrow n_+ + y_i$
14:     **end for**
15:     **for** $k \leftarrow 2 \ldots s$ **do**                                         $\triangleright$ find prefix sum
16:         $b_k \leftarrow b_k + b_{k-1}$
17:         $c_k \leftarrow c_k + c_{k-1}$
18:     **end for**
19:     $I^\star \leftarrow \infty$
20:     $w_a^\star \leftarrow 0$
21:     **for** $k \leftarrow 1 \ldots s$ **do**
22:         $I_k \leftarrow I(b_k, c_k, n - b_k, n_+ - c_k)$
23:         $\triangleright$ $I(|S_{u_l}|, |S_{u_l,+}|, |S_{u_r}|, |S_{u_r,+}|)$ returns the empirical critierion score for split $(a, w_{a,k})$ in $\mathcal{O}(1)$
    using either Gini index $I_G$ (this work) or Shannon's entropy $I_E$.
24:         **if** $I_k < I^\star$ **then**
25:             $I^\star \leftarrow I_k$
26:             $w_a^\star \leftarrow w_{a,k}$
27:         **end if**
28:     **end for**
29:     **return** $(a, w_a^\star)$
30: **end procedure**

---

$\square$

Given $q < 1$, the proportion of trees each sample is assigned to, $T$, the number of trees in the forest, $d_{\max}$, the maximum depth of each tree, $p$, the number of candidate attributes, $s$, the number of candidate splits for each attribute (usually $s \leq 30$), and $n = |S|$, size of the training set, we now prove the following:

**Theorem 2.** *Training* DYNFRS *yields a time complexity of* $\mathcal{O}(qTd_{\max}pn \log s)$.

*Proof.* For certain tree and a specific node $u$, we find the best split among $p$ randomly selected attributes $a_{1\cdots p}$, and we call FINDATTRIBUTEBESTSPLIT$(S_u, a_i, s)$ (Algorithm 3) $p$ times for each $i \in [p]$. From Lemma 1, finding the best split for the node $u$ has a time complexity of $\mathcal{O}(p|S_u| \log s)$. Then, summing $|S_u|$ over all tree nodes $u$ on that tree, we have $\sum_u |S_u| \leq d_{\max} qn$, since the root of the tree contains about $\lceil qT \rceil n/T \approx qn$ samples, and each layer has at most the same amount of samples as the root (layer 0). Therefore, the time complexity for training one DYNFRS tree can be bounded by $\mathcal{O}(d_{\max} qn \log s)$. Since there are $T$ independent trees in the forest, the time complexity for training a DYNFRS forest is $\mathcal{O}(qTd_{\max}pn \log s)$.

$\square$

**Theorem 3.** *Modification (sample addition or removal) in* DYNFRS *yields a time complexity of* $\mathcal{O}(qTd_{\max}ps)$ *if no attribute range changes occurs while* $\mathcal{O}(qTd_{\max}ps + cn_{\mathit{aff}} \log s)$ *otherwise (where $c$ denotes the number of attributes affected, and $n_{\mathit{aff}}$ denotes the sum of sample size among all affected nodes met by this modification request).*

*Proof.* When no attribute range change occurs on each tree, the modification request traverses a path from the root to a leaf with at most $d_{\max}$ nodes. For each node, we need to recalculate all the empirical criterion scores for all candidate splits $\mathcal{O}(ps)$. Since $\text{OCC}(q)$ guarantees that only $\lceil qT \rceil$ trees are affected by the modification requests, at most $qTd_{\max}$ nodes need the recalculation. So, the time complexity for one modification request yields $\mathcal{O}(qTd_{\max}ps)$.

When an attribute range occurs on $u$, it is necessary to call FINDATTRIBUTEBESTSPLIT$(S_u, a, s)$ for $u$ and the affected attribute $a$. Given that the affected nodes' sample sizes sum up to $n_{\text{aff}}$, and for each affected node, we need to resample at most $c \leq p$ attributes, and then Lemma 1 entails that the time complexity for completing all resampling is an additional $\mathcal{O}(cn_{\text{aff}} \log s)$.

$\square$

**Theorem 4.** *Query in* DYNFRS *yields a time complexity of* $\mathcal{O}(Td_{\max})$ *if no lazy tag is met, while* $\mathcal{O}(Td_{\max} + pn_{lzy} \log s)$ *otherwise (where $n_{lzy}$ denotes the sum of sample size among all nodes with lazy tag and met by this query).*

*Proof.* On each tree, the query starts with the root and ends at a leaf node, traversing a tree path with at most $d_{\max}$ nodes, and the query on DYNFRS aggregates the results of all $T$ trees, therefore querying without bumping into a lazy tag yields a time complexity of $\mathcal{O}(Td_{\max})$.

However, if the query reaches on a tagged node $u$, we need to perform a split on it, and by the proof of Theorem 2 and Lemma 1, finding the best split of node $u$ calls function FINDATTRIBUTEBESTSPLIT$(S_u, \cdot, s)$ $p$ times and results in a time complexity of $\mathcal{O}(p|S_u| \log s)$. As $n_{\text{lzy}}$ denotes the sum of sample sizes of all nodes with lazy tags met by the query, handling these lazy tags requires an additional time complexity of $\mathcal{O}(pn_{\text{lzy}} \log s)$.

$\square$

### A.3 IMPLEMENTATION

All of the experiments are conducted on a machine with AMD EPYC 9754 128-core CPU and 512 GB RAM in a Linux environment (Ubuntu 22.04.4 LTS), and all codes of DYNFRS are written in C++ and compiled with the g++ 11.4.0 compiler and the -O3 optimization flag enabled. To guarantee fair comparison, all tests are run on a single thread and are repeated 5 times with the mean and standard deviation reported.

DYNFRS is tuned using 5-fold cross-validation for each dataset, and the following hyperparameters are tuned using a grid search: Number of trees in the forest $T \in \{100, 150, 250\}$, maximum depth of each tree $d_{\max} \in \{10, 15, 20, 25, 30, 40\}$, and the number of sampled splits $s \in \{5, 15, 20, 30, 40\}$.

### A.4 BASELINES

HedgeCut and OnlineBoosting can not process real continuous input. Thus, all numerical attributes are discretized into 16 bins, as suggested in their works (Schelter et al., 2021; Lin et al., 2023). Both of them are not capable of processing samples with sparse attributes, so one-hot encoding is disabled for them. Additionally, it is impossible to train Hedgecut on datasets Synthetic and Higgs in our setting due to its implementation issue, as its complexity degenerates to $\mathcal{O}(pn^2)$ sometimes and consumes more than 256 GB RAM during training.

### A.5 DATASETS

**Purchase** (Sakar and Kastro, 2018; Dua and Graff, 2019) is primarily used to predict online shopping intentions, i.e., users' determination to complete a transaction. The dataset was collected from an online bookstore built on an osCommerce platform.

**Vaccine** (Bull et al., 2016; DrivenData, 2019) comes from data-mining competition in Driven-Data. It contains 26,707 survey responses, which were collected between October 2009 and June 2010. The survey asked 36 behavioral and personal questions. We aim to determine whether a person received a seasonal flu vaccine.

**Adult** (Becker and Kohavi, 1996; Dua and Graff, 2019) is extracted from the 1994 Census database by Barry Becker, and is used for predicting whether someone's income level is more than 50,000 dollars per year or not.

**Bank** (Moro et al., 2014; Dua and Graff, 2019) is related to direct marketing campaigns of a Portuguese banking institution dated from May 2008 to November 2010. The goal is to predict if the client will subscribe to a term deposit based on phone surveys.

**Heart** (Kaggle, 2018) is provided by Ulianova, and contains 70,000 patient records about cardiovascular diseases, with the label denoting the presence of heart disease.

**Diabetes** (Strack et al., 2014; Dua and Graff, 2019) encompasses a decade (1999-2008) of clinical diabetes records from 130 hospitals across the U.S., covering laboratory results, medications, and hospital stays. The goal is to predict whether a patient will be readmitted within 30 days of discharge.

**Synthetic** (Kaggle, 2016) focuses on the patient's appointment information, such as date, number of SMS sent, and alcoholism, aiming to predict whether the patient will show up after making an appointment.

**Higgs** (Baldi et al., 2014; Dua and Graff, 2019) consists of $1.1 \times 10^7$ signals characterized by 22 kinematic properties measured by detectors in a particle accelerator and 7 derived attributes. The goal is to distinguish between a background signal and a Higgs boson signal.

A.6 RESULTS

In this section, Table 3 presents the training time for each model, with OnlineBoosting being the fastest in most datasets while DYNFRS ranks first among Random Forest based methods. Table 4, 5, 6, and 7 despicts the runtime for model simultaneously unlearning 1, 10, 100 instances or $0.1\%$ and $1\%$ of all samples, where DYNFRS consistently outperforms all others in all settings and all datasets.

Table 3: Training time ($\downarrow$) of each model, measured in seconds (s), and the standard deviation is given with the same unit in a smaller font. "/" means the model is unable to train on that dataset.

| Datasets | DaRE | HedgeCut | Online Boosting | DYNFRS ($q = 0.1$) | DYNFRS ($q = 0.2$) |
|---|---|---|---|---|---|
| Purchase | $3.10_{\pm 0.0}$ | $1.05_{\pm 0.0}$ | $\mathbf{0.27}_{\pm 0.0}$ | $0.38_{\pm 0.0}$ | $0.72_{\pm 0.0}$ |
| Vaccine | $4.78_{\pm 0.0}$ | $431_{\pm 14}$ | $\mathbf{1.05}_{\pm 0.0}$ | $1.12_{\pm 0.0}$ | $2.27_{\pm 0.0}$ |
| Adult | $5.02_{\pm 0.1}$ | $11.8_{\pm 0.5}$ | $0.77_{\pm 0.0}$ | $\mathbf{0.61}_{\pm 0.0}$ | $1.15_{\pm 0.0}$ |
| Bank | $8.26_{\pm 0.2}$ | $8.44_{\pm 0.3}$ | $\mathbf{0.92}_{\pm 0.0}$ | $1.15_{\pm 0.0}$ | $2.37_{\pm 0.0}$ |
| Heart | $12.1_{\pm 0.2}$ | $3.51_{\pm 0.0}$ | $\mathbf{1.02}_{\pm 0.0}$ | $1.04_{\pm 0.0}$ | $1.96_{\pm 0.0}$ |
| Diabetes | $123_{\pm 1.0}$ | $162_{\pm 3.3}$ | $\mathbf{3.51}_{\pm 0.0}$ | $8.67_{\pm 0.0}$ | $18.2_{\pm 0.0}$ |
| NoShow | $65.4_{\pm 0.4}$ | $28.1_{\pm 0.3}$ | $\mathbf{1.68}_{\pm 0.0}$ | $3.08_{\pm 0.0}$ | $6.10_{\pm 0.0}$ |
| Synthetic | $1334_{\pm 6.3}$ | / | $\mathbf{40.7}_{\pm 0.9}$ | $66.3_{\pm 0.2}$ | $128_{\pm 0.4}$ |
| Higgs | $10793_{\pm 48}$ | / | $\mathbf{460}_{\pm 13}$ | $548_{\pm 1.2}$ | $1120_{\pm 1.0}$ |

Table 4: Runtime ($\downarrow$) for each model to unlearn 1 sample measured in milliseconds (ms), and the standard deviation is given with the same unit in a smaller font. "/" means the model is unable to train on that dataset or unlearning takes too long.

| Datasets | DaRE | HedgeCut | Online Boosting | DYNFRS |
|---|---|---|---|---|
| Purchase | $35.0_{\pm 15}$ | $1245_{\pm 343}$ | $83.4_{\pm 9.0}$ | $\mathbf{0.40}_{\pm 0.2}$ |
| Vaccine | $16.0_{\pm 15}$ | $33445_{\pm 14372}$ | $222_{\pm 53}$ | $\mathbf{1.40}_{\pm 0.9}$ |
| Adult | $10.6_{\pm 5.0}$ | $3596_{\pm 2396}$ | $249_{\pm 62}$ | $\mathbf{1.10}_{\pm 1.5}$ |
| Bank | $33.2_{\pm 17}$ | $2760_{\pm 371}$ | $227_{\pm 7.4}$ | $\mathbf{2.40}_{\pm 3.7}$ |
| Heart | $16.8_{\pm 10}$ | $972_{\pm 154}$ | $411_{\pm 13}$ | $\mathbf{0.50}_{\pm 0.2}$ |
| Diabetes | $293_{\pm 168}$ | $27654_{\pm 10969}$ | $753_{\pm 143}$ | $\mathbf{7.30}_{\pm 5.4}$ |
| NoShow | $330_{\pm 176}$ | $1243_{\pm 94}$ | $570_{\pm 69}$ | $\mathbf{0.30}_{\pm 0.0}$ |
| Synthetic | $2265_{\pm 3523}$ | / | $5225_{\pm 241}$ | $\mathbf{2.50}_{\pm 3.7}$ |
| Higgs | $174_{\pm 135}$ | / | $73832_{\pm 4155}$ | $\mathbf{1.60}_{\pm 1.8}$ |

Table 5: Runtime (↓) for each model to unlearn 10 samples measured in milliseconds (ms), and the standard deviation is given with the same unit in a smaller font. "/" means the model is unable to train on that dataset or unlearning takes too long.

| Datasets | DaRE | HedgeCut | Online Boosting | DYNFRS |
|---|---|---|---|---|
| Purchase | $295_{\pm 54}$ | $10973_{\pm 643}$ | $183_{\pm 21}$ | $\mathbf{12.3}_{\pm 4.8}$ |
| Vaccine | $285_{\pm 160}$ | $222333_{\pm 64756}$ | $418_{\pm 41}$ | $\mathbf{9.20}_{\pm 2.8}$ |
| Adult | $148_{\pm 79}$ | $51831_{\pm 2795}$ | $389_{\pm 48}$ | $\mathbf{4.80}_{\pm 2.4}$ |
| Bank | $320_{\pm 108}$ | $18091_{\pm 1524}$ | $423_{\pm 47}$ | $\mathbf{7.6}_{\pm 2.7}$ |
| Heart | $162_{\pm 46}$ | $5524_{\pm 335}$ | $625_{\pm 36}$ | $\mathbf{6.80}_{\pm 2.8}$ |
| Diabetes | $2773_{\pm 877}$ | $211640_{\pm 79652}$ | $1096_{\pm 172}$ | $\mathbf{85.5}_{\pm 38}$ |
| NoShow | $2217_{\pm 723}$ | $17235_{\pm 17905}$ | $712_{\pm 74}$ | $\mathbf{18.2}_{\pm 123}$ |
| Synthetic | $92279_{\pm 42634}$ | / | $6015_{\pm 301}$ | $\mathbf{77.3}_{\pm 34}$ |
| Higgs | $20119_{\pm 31897}$ | / | $104063_{\pm 1258}$ | $\mathbf{32.1}_{\pm 8.6}$ |

Table 6: Runtime (↓) for each model to unlearn 100 samples measured in milliseconds (ms), and the standard deviation is given with the same unit in a smaller font. "/" means the model is unable to train on that dataset or unlearning takes too long.

| Datasets | DaRE | HedgeCut | Online Boosting | DYNFRS |
|---|---|---|---|---|
| Purchase | $3591_{\pm 186}$ | $83649_{\pm 2047}$ | $275_{\pm 12}$ | $\mathbf{70.4}_{\pm 11}$ |
| Vaccine | $2385_{\pm 408}$ | $1703355_{\pm 157274}$ | $792_{\pm 15.06}$ | $\mathbf{82.6}_{\pm 11}$ |
| Adult | $954_{\pm 158}$ | $219392_{\pm 34630}$ | $632_{\pm 24}$ | $\mathbf{32.2}_{\pm 4.0}$ |
| Bank | $3546_{\pm 302}$ | $195014_{\pm 15854}$ | $740_{\pm 20}$ | $\mathbf{78.8}_{\pm 13}$ |
| Heart | $1502_{\pm 543}$ | $33806_{\pm 5385}$ | $986_{\pm 75}$ | $\mathbf{59.3}_{\pm 11}$ |
| Diabetes | $23833_{\pm 10330}$ | / | $2071_{\pm 115}$ | $\mathbf{578}_{\pm 50}$ |
| NoShow | $23856_{\pm 3978}$ | $57021_{\pm 6327}$ | $1120_{\pm 92}$ | $\mathbf{117}_{\pm 10}$ |
| Synthetic | $1073356_{\pm 420053}$ | / | $7609_{\pm 171}$ | $\mathbf{889}_{\pm 287}$ |
| Higgs | $165122_{\pm 149092}$ | / | $145386_{\pm 5677}$ | $\mathbf{642}_{\pm 317}$ |

Table 7: Left: runtime (↓) for unlearning $0.1\%$ of the training set between models. Right: runtime (↓) for unlearning $1\%$ of the training set between models. Each cell contains the elapsed time in seconds (s), and the standard deviation is given with the same unit in a smaller font. "/" means the model is unable to train on that dataset or unlearning takes too long.

| Datasets | DaRE | HedgeCut | Online Boosting | DYNFRS | DaRE | HedgeCut | Online Boosting | DYNFRS |
|---|---|---|---|---|---|---|---|---|
| Purchase | $0.35_{\pm 0.1}$ | $11.25_{\pm 1.5}$ | $0.17_{\pm 0.0}$ | $\mathbf{0.01}_{\pm 0.0}$ | $3.39_{\pm 0.7}$ | $76.0_{\pm 3.0}$ | $0.28_{\pm 0.0}$ | $\mathbf{0.07}_{\pm 0.0}$ |
| Vaccine | $0.47_{\pm 0.1}$ | $404.73_{\pm 69}$ | $0.61_{\pm 0.1}$ | $\mathbf{0.02}_{\pm 0.0}$ | $5.01_{\pm 1.0}$ | $4054_{\pm 427}$ | $0.98_{\pm 0.0}$ | $\mathbf{0.13}_{\pm 0.0}$ |
| Adult | $0.44_{\pm 0.3}$ | $88.1_{\pm 27}$ | $0.49_{\pm 0.1}$ | $\mathbf{0.01}_{\pm 0.0}$ | $3.39_{\pm 0.6}$ | $516_{\pm 23}$ | $0.80_{\pm 0.0}$ | $\mathbf{0.09}_{\pm 0.0}$ |
| Bank | $1.15_{\pm 0.3}$ | $47.3_{\pm 6.0}$ | $0.61_{\pm 0.1}$ | $\mathbf{0.02}_{\pm 0.0}$ | $14.7_{\pm 2.3}$ | $418_{\pm 25}$ | $0.96_{\pm 0.0}$ | $\mathbf{0.16}_{\pm 0.0}$ |
| Heart | $0.70_{\pm 0.2}$ | $20.0_{\pm 1.7}$ | $0.85_{\pm 0.0}$ | $\mathbf{0.03}_{\pm 0.0}$ | $8.43_{\pm 1.2}$ | $145_{\pm 10}$ | $1.23_{\pm 0.0}$ | $\mathbf{0.20}_{\pm 0.0}$ |
| Diabetes | $23.8_{\pm 2.7}$ | $694_{\pm 61}$ | $2.12_{\pm 0.1}$ | $\mathbf{0.57}_{\pm 0.1}$ | $258_{\pm 20}$ | / | $3.51_{\pm 0.1}$ | $\mathbf{2.50}_{\pm 0.1}$ |
| NoShow | $18.8_{\pm 2.7}$ | $57.0_{\pm 6.3}$ | $1.10_{\pm 0.1}$ | $\mathbf{0.10}_{\pm 0.0}$ | $268_{\pm 7.5}$ | / | $1.90_{\pm 0.1}$ | $\mathbf{0.56}_{\pm 0.0}$ |
| Synthetic | $10790_{\pm 5348}$ | / | $13.1_{\pm 0.6}$ | $\mathbf{5.68}_{\pm 0.2}$ | / | / | $44.2_{\pm 1.4}$ | $\mathbf{27.4}_{\pm 0.9}$ |
| Higgs | / | / | $188_{\pm 7.1}$ | $\mathbf{39.2}_{\pm 0.7}$ | / | / | $456_{\pm 4.7}$ | $\mathbf{201}_{\pm 9.6}$ |

## A.7 SPACE COMPLEXITY AND MEMORY CONSUMPTION

The space complexity of DYNFRS is $\mathcal{O}(qTn + Tvps)$ where $T$ is the number of trees in the forest, $n$ is the number of samples in the training sets, $q$ is the factor used in OCC, $v$ is the average number of nodes in each tree, $p$ is the number of attributes considered by each node, and $s$ is the number of candidate splits. Since we store training samples on each leaf, and each tree occupies $qn$ samples on

average, then the leaf statistics sums up to $\mathcal{O}(qTn)$. As mentioned in section 4.3, we store an extra $\mathcal{O}(ps)$ split statistics on each node so that these split statistics contribute up to $\mathcal{O}(Tvps)$ space.

We compare the maximum resident space of DYNFRS and DaRE for building on the training set. We found that DaRE, which has a space complexity of $\mathcal{O}(Tn+Tvps)$, has larger memory consumption than DYNFRS in all datasets. We use /usr/bin/time in Linux to evaluate the maximum resident set size.

Table 8: The maximum resident set size ($\downarrow$) of each model measured in megabytes with standard deviation shown in a smaller font.

| Datasets | DaRE | DYNFRS |
|---|---|---|
| Purchase | $398.4_{\pm1.4}$ | $\mathbf{83.2}_{\pm1.6}$ |
| Vaccine | $782.2_{\pm1.6}$ | $\mathbf{492.8}_{\pm1.6}$ |
| Adult | $460.6_{\pm2.2}$ | $\mathbf{232.0}_{\pm0}$ |
| Bank | $801.0_{\pm1.8}$ | $\mathbf{300.0}_{\pm0}$ |
| Heart | $254.4_{\pm1.7}$ | $\mathbf{252.2}_{\pm1.6}$ |
| Diabetes | $7148_{\pm39}$ | $\mathbf{2512}_{\pm7.6}$ |
| NoShow | $3792_{\pm19}$ | $\mathbf{761.6}_{\pm2.0}$ |
| Synthetic | $8526_{\pm65}$ | $\mathbf{5827}_{\pm7.8}$ |
| Higgs | $60528_{\pm789}$ | $\mathbf{58263}_{\pm52}$ |

## A.8 MULTI-CLASS CLASSIFICATION

DYNFRS is capable of handling multi-class classification tasks since $\text{OCC}(q)$ and LZY do not affect the functionality of the forest. We tested DYNFRS's prediction performance and unlearning boost on 3 datasets — Optical (Hyafil and Rivest, 1976), Pen (Alpaydin and Alimoglu, 1996), and Letter (Slate, 1991). Unfortunately, existing random forest unlearning methods (DaRE and Hedge-Cut) have no implementation for multi-class classification, so we only include the Random Forest Classifier implementation (we call it Vanilla in the following) in scikit-learn as our baseline.

Table 9: Datasets specifications. (# train: number of training samples; # test: number of testing samples; # attr: number of attributes; # class: number of label classes.)

| Datasets | # train | # test | # attr | # class |
|---|---|---|---|---|
| Optical | 3823 | 1797 | 64 | 10 |
| Pen | 7494 | 3498 | 16 | 10 |
| Letter | 15000 | 5000 | 16 | 26 |

Table 10: Each model's predictive performance, training time, and unlearning boost with standard deviation shown in a smaller font.

| | Accuracy ($\uparrow$) | | Train Time ($\downarrow$,ms) | | Unlearn Boost ($\uparrow$) | |
|---|---|---|---|---|---|---|
| Datasets | Vanilla | DYNFRS | Vanilla | DYNFRS | Vanilla | DYNFRS |
| Optical | $.9694_{\pm.002}$ | $\mathbf{.9707}_{\pm.002}$ | $585.4_{\pm1.9}$ | $\mathbf{445.4}_{\pm2.7}$ | $1_{\pm0}$ | $\mathbf{292.5}_{\pm12.1}$ |
| Pen | $.9649_{\pm.002}$ | $\mathbf{.9696}_{\pm.002}$ | $996.2_{\pm1.7}$ | $\mathbf{813.2}_{\pm8.0}$ | $1_{\pm0}$ | $\mathbf{603.0}_{\pm44.8}$ |
| Letter | $.9603_{\pm.002}$ | $\mathbf{.9624}_{\pm.001}$ | $1666_{\pm7.6}$ | $\mathbf{1530}_{\pm20}$ | $1_{\pm0}$ | $\mathbf{1006}_{\pm54.2}$ |

Results show that DYNFRS outperforms the vanilla Random Forest in terms of predictive performance and training time for all datasets. Additionally, DYNFRS still shows splendid unlearning efficiency on these datasets. In all datasets, DYNFRS outperforms Vanilla by 2-3 order of magnitudes in terms of unlearning efficiency. Also notice that DYNFRS is poorly optimized for multi-class classification tasks.

In the multi-class classification setting, we suggest picking $q = 0.5$ for $\text{OCC}(q)$, because the rise in class number leads to the drop of sample size to each class (roughly $n/c$, where $n$ is the number of training samples and $c$ is the number of classes) and rising $q$ enable each tree is accessible to more samples belonging to a specific class and lead to better predictive performance eventually. For all datasets, we set the hyperparameters of DYNFRS to $T = 100$, $d_{\max} = 20$, and $s = 1$.

## A.9 REGRESSION

Regression tasks are not implemented and evaluated by any of the existing tree-based unlearning methods. Since there is no such baseline, we compare DYNFRS's predictive performance (via mean squared error) and unlearning efficiency (via unlearning boost) against the naive retraining method. We used a large and popular dataset Bike (Fanaee-T, 2013) for the regression task.

Table 11: Datasets specifications. (# train: number of training samples; # test: number of testing samples; # attr: number of attributes.)

| Datasets | # train | # test | # attr |
|---|---|---|---|
| Bike | 13903 | 3476 | 16 |

Table 12: Each model's predictive performance, training time, and unlearning boost with standard deviation are shown in a smaller font.

| | MSE ($\downarrow$) | | Train Time ($\downarrow$,ms) | | Unlearn Boost ($\uparrow$) | |
|---|---|---|---|---|---|---|
| Datasets | Vanilla | DYNFRS | Vanilla | DYNFRS | Vanilla | DYNFRS |
| Bike | $3.904_{\pm.071}$ | $\mathbf{3.011}_{\pm.167}$ | $6796_{\pm6.5}$ | $\mathbf{3412}_{\pm54}$ | $1_{\pm0}$ | $\mathbf{2368}_{\pm118}$ |

Results show that DYNFRS outperforms the vanilla Random Forest in terms of predictive performance and training time. Note that the implementation of DYNFRS regressor is rough and poorly optimized due to short time slots. However, DYNFRS still shows outstanding unlearning efficiency on regression task, as DYNFRS achieves an averaged 2368 unlearning boost in dataset Bike.

In the regression setting, we suggest picking $q = 0.5$ for lower mean squared error. In the experiment, we set the hyperparameters of DYNFRS to $T = 100$, $d_{\max} = 15$, and $s = 5$.

## A.10 ONLINE MIXED DATA STREAM

Table 13: DYNFRS's latency ($\downarrow$) of sample addition/removal and querying in 4 scenarios measured in microseconds ($\mu$s). # add/del/qry: the number of sample addition/removal/querying requests; add/del/qry lat.: the latency of sample addition/removal/querying. Each cell contains the averaged latency with its minimum and maximum values listed in a smaller font.

| No. | # add | # del | # qry | add lat. ($\downarrow$, $\mu$s) | del lat. ($\downarrow$, $\mu$s) | qry lat. ($\downarrow$, $\mu$s) |
|---|---|---|---|---|---|---|
| 1 | $5 \cdot 10^5$ | $5 \cdot 10^5$ | $10^6$ | $406.2$ [150, 450894] | $437.6$ [134, 394801] | $3680$ [209, 1885030] |
| 2 | $5 \cdot 10^5$ | $5 \cdot 10^5$ | $10^6$ | $122.7$ [30, 168984] | $120.2$ [25, 146300] | $1218$ [23, 1026964] |
| 3 | $5 \cdot 10^4$ | $5 \cdot 10^4$ | $10^6$ | $140.0$ [30, 81661] | $139.2$ [32, 101429] | $299.5$ [21, 861151] |
| 4 | $5 \cdot 10^3$ | $5 \cdot 10^3$ | $10^6$ | $145.5$ [44, 55954] | $140.5$ [38, 29810] | $72.3$ [19, 212915] |

To simulate a large-scale database, we use the Higgs dataset, the largest in our study. We train DYNFRS on $88,000,000$ samples and feed it with mixed data streams with different proportions of modification requests. Scenario 1 is the vanilla single-thread setting, while scenarios 2, 3, and 4 employ 25 threads using OpenMP. DYNFRS achieves an averaged latency of less than 0.15 ms for modification requests (Table 13 column # add and # del) and significantly outperforms DaRE, which requires 180 ms to unlearn a single instance on average. Query latency drops from 1.2 ms to 0.07 ms as the number of modification requests declines, as fewer lazy tags are introduced to trees.

These results are striking: while it takes over an hour to train a vanilla Random Forest on Higgs, DYNFRS maintains exceptionally low latency that is measured in $\mu$s, even in the single-threaded setting. This makes DYNFRS highly suited for real-world scenarios, especially when querying constitutes a large proportion of requests (Table 13 Scenario 4).

## A.11 EFFECTS ON NUMBER OF CANDIDATES

We assessed the predictive performance of DYNFRS with differnet candidate number $s$. We tested DYNFRS($q = 0.1$) with $s \in \{1, 5, 10, 20, 30, 50\}$ and report the predictive performance on all the binary classification datasets (same settings as in Section 5.2). The results are summarized in the table below:

Table 14: The predictive performance ($\uparrow$) of DYNFRS($q = 0.1$) with $s \in \{1, 5, 10, 20, 30, 50\}$.

| Datasets | $s = 1$ | $s = 5$ | $s = 10$ | $s = 20$ | $s = 30$ | $s = 50$ |
|---|---|---|---|---|---|---|
| Purchase | $.9242_{\pm.001}$ | $.9313_{\pm.000}$ | $.9323_{\pm.000}$ | $.9329_{\pm.001}$ | $.9328_{\pm.001}$ | $.9330_{\pm.001}$ |
| Vaccine | $.7911_{\pm.002}$ | $.7910_{\pm.000}$ | $.7908_{\pm.001}$ | $.7910_{\pm.002}$ | $.7911_{\pm.002}$ | $.7912_{\pm.001}$ |
| Adult | $.8558_{\pm.001}$ | $.8610_{\pm.001}$ | $.8624_{\pm.001}$ | $.8635_{\pm.001}$ | $.8635_{\pm.000}$ | $.8640_{\pm.001}$ |
| Bank | $.9323_{\pm.000}$ | $.9399_{\pm.000}$ | $.9409_{\pm.000}$ | $.9414_{\pm.001}$ | $.9417_{\pm.000}$ | $.9417_{\pm.001}$ |
| Heart | $.7365_{\pm.001}$ | $.7359_{\pm.001}$ | $.7357_{\pm.001}$ | $.7359_{\pm.002}$ | $.7357_{\pm.000}$ | $.7351_{\pm.001}$ |
| Diabetes | $.6429_{\pm.001}$ | $.6453_{\pm.001}$ | $.6451_{\pm.001}$ | $.6446_{\pm.001}$ | $.6442_{\pm.001}$ | $.6443_{\pm.001}$ |
| NoShow | $.7278_{\pm.001}$ | $.7332_{\pm.000}$ | $.7328_{\pm.000}$ | $.7332_{\pm.000}$ | $.7323_{\pm.001}$ | $.7328_{\pm.000}$ |
| Synthetic | $.9352_{\pm.000}$ | $.9415_{\pm.000}$ | $.9421_{\pm.000}$ | $.9422_{\pm.000}$ | $.9424_{\pm.000}$ | $.9423_{\pm.000}$ |
| Higgs | $.7277_{\pm.000}$ | $.7409_{\pm.000}$ | $.7423_{\pm.000}$ | $.7431_{\pm.000}$ | $.7431_{\pm.000}$ | $.7431_{\pm.000}$ |

From the result, we find that the performance peaks around $s = 20$ in most datasets, and the result of $s = 30$ and $s = 50$ has no significant difference. However, in datasets Heart, Diabetes, and NoShow, a smaller $s$ has an even higher predictive performance, suggesting that considering more candidates might not always be the best choice. These results indicate that the optimal $s$ is dataset-specific and can be tuned for improved predictive performance in DYNFRS.

## A.12 HYPERPARAMETERS

All hyperparameters of DYNFRS are listed in Table 15. Specially, we set the minimum split size of each node to be 10 for all datasets.

Table 15: Hyperparameters used by DYNFRS with both $q = 0.1$ and $q = 0.2$ setting.

| Datasets | $T$ | $d_{\max}$ | $s$ |
|---|---|---|---|
| Purchase | 250 | 10 | 30 |
| Vaccine | 250 | 20 | 5 |
| Adult | 100 | 20 | 30 |
| Bank | 250 | 25 | 30 |
| Heart | 150 | 15 | 5 |
| Diabetes | 250 | 30 | 5 |
| NoShow | 250 | 20 | 5 |
| Synthetic | 150 | 40 | 30 |
| Higgs | 100 | 30 | 20 |

