# OpenReview forum: "DynFrs: An Efficient Framework for Machine Unlearning in Random Forest"
_ICLR.cc/2025/Conference — ICLR 2025 Poster_

### Official Review · Reviewer_THi8 · 2024-10-29

**Soundness:** 2
**Presentation:** 2
**Contribution:** 2
**Rating:** 5
**Confidence:** 3

**Summary:**

This paper addresses the unlearning problem in Random Forests for privacy concerns, focusing on efficient data point removal or addition. The DYNFRS framework is proposed for efficient machine unlearning in Random Forests while maintaining predictive accuracy.

**Strengths:**

1. The paper presents a comprehensive set of experiments, encompassing various types of unlearning scenarios.

2. The proposed method, DYNFRS, demonstrates significant efficiency improvements compared to existing baselines.

**Weaknesses:**

1. My primary concern is the limited novelty of this paper. DYNFRS appears to be an assemblage of components with minor modifications. The framework consists of three main components: OCC which performs subsampling on trees instead of training samples, essentially randomly selecting $k$ trees; LZY which utilizes the lazy tag concept and  replaces subtree retraining with reconstructing the tree path that connects the modified node u and a leaf; ERT which  is an existing method, with modifications like  only s candidates on one attribute instead of all possible splits. DYNFRS, therefore, seems to be a combination of existing techniques with relatively minor alterations. While the integration of these components may offer some benefits, it is questionable whether the contribution of these improvements meets the standard expected at ICLR.

2. The clarity of the paper's writing could be further improved:

- It would be better if some redundant theorems and formulas could be avoided. For instance:

  (a) Theorem 1's conclusion is trivial, especially given the assumptions of sample independence and the splitting characteristics of tree models.

    (b) Lemma 1 is also trivial, since the major difference in the complexity stems from the introduction of $s$ candidates compared to classic methods.

  If the authors could more directly and concisely highlight these key points instead of using extensive formulas and theorems, it may be further enhance the clarity of the paper.

- Section 4.2 requires a necessary introduction to the lazy tag, particularly since lazy tagging is a classic technique in segment tree problems.

- The sources of observations in Line 209 and Line 260 are unclear. These assertions need proper citations to relevant literature. For example, in point (2) in Line 260, there are considerable research demonstrating the existence of a small portion of neurons or attention heads in neural networks that also can significantly influence inference outputs during the inference phase [1,2,3].

3. The experimental section focuses solely on binary classification, neglecting multi-class scenarios. It would be beneficial to consider multi-class problems.

4. *OCC(0.2) sometimes improves the forest's predictive performance.*  Could the authors provide some insights  for  latent reasons?

5. In section 4.3, introducing consideration of only $s$ candidates, although has a higher chance to remain unchanged when adding or removing data points, it is also entirely possible to not obtain the best split in all possible splits. The authors should discuss this additional loss caused by only $s$ candidates.

[1] Morcos, A. S., Barrett, D. G., Rabinowitz, N. C., & Botvinick, M. (2018, February). On the importance of single directions for generalization. In *International Conference on Learning Representations*.

[2] Goh, Gabriel, Nick Cammarata, Chelsea Voss, Shan Carter, Michael Petrov, Ludwig Schubert, Alec Radford, and Chris Olah. "Multimodal neurons in artificial neural networks." *Distill* 6, no. 3 (2021): e30.

[3] Zhu, W., Zhang, Z., & Wang, Y. Language Models Represent Beliefs of Self and Others. In *Forty-first International Conference on Machine Learning*.

**Questions:**

See my Weaknesses.

---

> ### Author Response · Authors · 2024-11-21
> **Reply to Reviewer THi8**
>
> We appreciate the reviewers' efforts to engage with our work in depth and their thoughtful questions, which have helped us strengthen our contributions. Here are the responses to your concerns.
>
> ### **1. My primary concern is the limited novelty of this paper.**
>
> Here are the main contributions that make DynFrs stand out.
> While simple and easy to implement, the OCC subsampling method is (to the best of our knowledge) the first approach to applying subsampling on trees, which benefits unlearning quite a lot. Secondly, LZY makes DynFrs the first tree-based unlearning framework capable of avoiding the need to retrain entire subtrees. We acknowledge that our choice of Extremely Randomized Trees was driven by their minimal dependency on training samples (which is suitable for unlearning). Ultimately, DynFrs may not be theoretical ground-breaking. Still, it is an attempt to make a tree-based model more applicable to real-world applications where sample addition and deletion happen rapidly. The goal of DynFrs is to fit tree-based models into mixed online data streams, allowing the forest to shrink and grow dynamically with the ever-changing real-world data.
>
> ### **2. The clarity of the paper's writing could be further improved.**
>
> We would thank the reviewer for writing the thorough and detailed guidelines for how to improve the manuscript's clarity. We will try to adjust our manuscript accordingly.
>
> ### **3. The experimental section focuses solely on binary classification, neglecting multi-class scenarios.**
>
> We have now modified a multi-class DynFrs that can handle multi-class classification. Please read the results in the section **2. Is DynFrs capable of conquering multi-class classification?** in public comment **Frequently Asked Questions**.
>
> ### **4. OCC(0.2) sometimes improves the forest's predictive performance. Could the authors provide some insights for latent reasons?**
>
> We believe that there are two latent reasons that account for the improvement in predictive performance due to OCC(0.2).
>
> First, OCC(0.2) brings more randomness into the whole forest. As claimed by Breiman [1], randomness is essential to the whole forest's accuracy. To quantify such "randomness", we claim that OCC(0.2) decreases the correlation between trees in the forest since each tree is trained on different small subsets of the whole training set. One can prove that
> $$ \mathbf{Var}(\Phi(X)) = \mathbf{Var}(\phi(X)) \left( \frac{1}{T} + \frac{T - 1}{T} \mathbf{Corr}\left( \phi_1(X), \phi_2(X) \right) \right) $$
> where $\Phi(X)$ is the output of the whole forest given input $X$, and $\phi(X)$ is the output of a single tree, and $\mathbf{Corr}\left( \phi_1(X), \phi_2(X) \right)$ is the correlation between two different trees in the forest.
>
> As OCC(0.2) decreases $\mathbf{Corr}\left( \phi_1(X), \phi_2(X) \right)$ by making each tree train on different subsets of the training set, it may potentially decrease $\mathbf{Var}(\Phi(X))$. The drop in variance might lead to less error in, at least, regression tasks. Through the bias-variance decomposition [2], the expected value for squared error loss is $\mathrm{noise} + \mathrm{bias}^2 + \mathrm{variance}$, where the $\mathrm{variance}$ stands for $\mathbf{Var}(\Phi(X))$.
>
> Second, OCC(0.2) might decrease the **pointwise hypothesis stability** $\beta$ [3], which is defined as
> $$ \forall i,\, 1 \le i \le |S|,\, \mathbb E_S\left[\left| \mathcal M_S(X_i) - \mathcal M_{S \backslash \{ (X_i, Y_i) \}}(X_i) \right|\right] \le \beta $$
> where $\mathcal M_S(X_i)$ is the output of a model trained on training set $S$ and has an input $X_i$. Intuitively, the stability of a model is how large its output will change when a random sample is removed (unlearned) from its training set.
>
> Suppose that a vanilla Random Forest in which each tree trains on the whole training set has a stability $\beta$, then we suggest that applying OCC(0.2) will have a stability $\beta'$ s.t. $\beta' < \beta$. Since removing one sample $(X_i, Y_i)$ only affects $\lceil qT \rceil$ trees instead of all trees, the contribution of the unaffected trees to $\left| \mathcal M(S, X_i) - \mathcal M(S \backslash \{ (X_i, Y_i) \}, X_i) \right|$ is zero. After applying OCC(0.2), we hypothesize that the stability $q' \approx q\beta$. Intuitively, The limitation that each sample appears in only $\lceil qT \rceil$ trees makes the whole trees more stable when one of the samples is removed, as the remaining $T - \lceil qT \rceil$ trees stay unchanged.

---

> ### Author Response · Authors · 2024-11-21
> **Reply to Reviewer THi8 (cont'd)**
>
> The benefit of having a smaller stability is that it would decrease the generalization error of the model with probability. Suppose that a model $\mathcal M_S$ has a pointwise hypothesis stability $\beta$ and a loss function $\mathcal L$ that is $L$-Lipchitz continuous and has a upper bound $M$, also denote $n = |S|$, then the following holds [3]:
> $$ \Pr \left[R(A_S) \le R_{\mathrm{emp}}(A_S) + \sqrt{\delta^{-1}} \sqrt{\frac{M^2}{2n} + 6M\beta} \right] \ge 1 - \delta $$
> where $R(A_S) \triangleq \mathbb E_{(X_i, Y_i) \sim \mathcal D}[\mathcal L(A_S(X_i), Y_i)]$ and $R_{\mathrm{emp}}(A_S) \triangleq \mathbb E_{(x_i, y_i) \sim S}[\mathcal L(A_S(x_i), y_i)]$. This probability inequality enables us to bound the generalization error from empirical error (the error estimated on the training set), and smaller stability makes the bound tighter. Smaller stability might be the key to why OCC(0.2) sometimes improves the forest's predictive performance.
>
> ### **5. In section 4.3, introducing consideration of only $s$ candidates, although it has a higher chance to remain unchanged when adding or removing data points, it is also entirely possible to not obtain the best split in all possible splits. The authors should discuss this additional loss caused by only $s$ candidates.**
>
> It is possible that none of the candidates is the best split among all possible splits. However, it is not true that relying on the best candidate gives the tree a higher accuracy. As finding the optimal decision tree is NP-Hard, deterministic searching methods [4] suggest that finding the split with the optimal empirical criterion score does not always produce the best-performing decision tree. So, not encapsulating the best split among all splits might not be a loss. Always greedily choosing the best split can lead to a good-performing Decision Tree, but selecting the second-best or a relatively good split brings more potential to the tree (think of the tree as not trapped in local minimum). Not choosing the best split may also prevent the tree from overfitting [5]. Also, the trade-offs are analyzed in the original work for Extremely Randomized Trees [6].
>
> **References**
>
> [1] L. Breiman, "Random Forests," Machine Learning, vol. 45, no. 1, pp. 5–32, Oct. 2001, doi: 10.1023/A:1010933404324.
>
> [2] G. Louppe, "Understanding Random Forests: From Theory to Practice," arXiv, 2015. Accessed: Jun. 23, 2024. [Online]. Available: http://arxiv.org/abs/1407.7502
>
> [3] O. Bousquet and A. Elisseeff, "Stability and Generalization," Journal of Machine Learning Research, vol. 2, no. Mar, pp. 499–526, 2002.
>
> [4] E. Demirović et al., "MurTree: Optimal Decision Trees via Dynamic Programming and Search," Journal of Machine Learning Research, vol. 23, no. 26, pp. 1–47, 2022.
>
> [5] L. Z. You, P. Y. Han, K. Z. B. Kamarudin, O. S. Yin, and H. F. San, "Bayesian Optimization Driven Strategy for Detecting Credit Card Fraud with Extremely Randomized Trees," MethodsX, p. 103055, Nov. 2024, doi: 10.1016/j.mex.2024.103055.
>
> [6] P. Geurts, D. Ernst, and L. Wehenkel, "Extremely randomized trees," Mach Learn, vol. 63, no. 1, pp. 3–42, Apr. 2006, doi: 10.1007/s10994-006-6226-1.

---

> ### Comment · Reviewer_THi8 · 2024-11-22
>
> Thank you for your reply. After reading your response, I acknowledge the effectiveness of the DYNFRS framework. However, I still have concerns regarding the novelty of the proposed framework (as mentioned in question 1). There are still some issues that need addressing in the manuscript, such as the lack of sufficient references for certain claims and the inclusion of trivial theorems (as mentioned in question 2). Additionally, some important ablation experiments (as noted in question 5) are still absent. Given these considerations, I maintain my score.
>
> However, I also acknowledge that I am not familiar with this specific field, which may have led to an inaccurate assessment of its innovativeness.

---

> ### Author Response · Authors · 2024-11-26
> **Follow-Up Reply to Reviewer THi8**
>
> Dear Reviewer THi8,
>
> Thank you for taking the time to provide your thoughtful feedback. We truly appreciate your acknowledgment of the effectiveness of the DynFrs framework and your detailed considerations regarding our manuscript.
>
> We have addressed the specific points you raised, including adding more references to substantiate our claims and incorporating the missing ablation experiments to provide a comprehensive evaluation.
>
> However, we would make the following clarifications:
>
> (1) The sentence
> > One observation is that if sample $\langle \mathbf x_{i}, y_{i} \rangle \in S$ does not occurs in tree $\varphi_t$ (i.e., $\langle \mathbf x_{i}, y_{i} \rangle \notin S^{(t)}$), then tree $\varphi_t$ is unaffected when unlearning $\langle \mathbf x_{i}, y_{i} \rangle$.
>
> is self-evident and does not require references because if a sample does not appear in the model, then we cannot *unlearn* it from the model because the model has never learned it. To fix it, we change the phrase *one observation* to *key idea*, emphasizing its role as a foundational concept rather than an empirical observation. Also, the observations in Section 4.2 are true for all Random Forests models and its variants, and they are *properties* of tree-based models. However, we still added the reference you suggested to enrich our manuscript. To be concise, we have changed the description in Section 4.2 from *observations on tree-based models* to *observations on differences between tree-based methods and neural network based methods*.
>
> (2) The inclusion of Lemma 1 and Theorem 1 is necessary. As Lemma 1 is the basis for Theorem 2, 3, 4 and it introduce a new fashion for finding the node's best split without sorting all the samples. On the other hand, Theorem 1 is the theoretical back up for DynFrs's unlearning exactness. This guarantees both the model's predictive performance after unlearning and its resistance to privacy leaks, ensuring the theoretical soundness and security of DynFrs.
>
> As you suggested, we ran an ablation experiment regarding the number of candidate split $s$. We tested DynFrs($q = 0.1$) with $s \in \\{1, 5, 10, 20, 30, 50\\}$ and report the predictive performance on all the binary classification datasets (same settings as in the manuscript). The results are summarized in the table below:
>
> |**Datasets**|$s=1$|$s=5$|$s=10$|$s=20$|$s=30$|$s=50$|
> |---|---|---|---|---|---|---|
> |**Purchase**|.9242$\pm.001$|.9313$\pm.000$|.9323$\pm.000$|.9329$\pm.001$|.9328$\pm.001$|.9330$\pm.001$|
> |**Vaccine**|.7911$\pm.002$|.7910$\pm.000$|.7908$\pm.001$|.7910$\pm.002$|.7911$\pm.002$|.7912$\pm.001$|
> |**Adult**|.8558$\pm.001$|.8610$\pm.001$|.8624$\pm.001$|.8635$\pm.001$|.8635$\pm.000$|.8640$\pm.001$|
> |**Bank**|.9323$\pm.000$|.9399$\pm.000$|.9409$\pm.000$|.9414$\pm.001$|.9417$\pm.000$|.9417$\pm.001$|
> |**Heart**|.7365$\pm.001$|.7359$\pm.001$|.7357$\pm.001$|.7359$\pm.002$|.7357$\pm.000$|.7351$\pm.001$|
> |**Diabetes**|.6429$\pm.001$|.6453$\pm.001$|.6451$\pm.001$|.6446$\pm.001$|.6442$\pm.001$|.6443$\pm.001$|
> |**NoShow**|.7278$\pm.001$|.7332$\pm.000$|.7328$\pm.000$|.7332$\pm.000$|.7323$\pm.001$|.7328$\pm.000$|
> |**Synthetic**|.9352$\pm.000$|.9415$\pm.000$|.9421$\pm.000$|.9422$\pm.000$|.9424$\pm.000$|.9423$\pm.000$|
> |**Higgs**|.7277$\pm.000$|.7409$\pm.000$|.7423$\pm.000$|.7431$\pm.000$|.7431$\pm.000$|.7431$\pm.000$|
>
> From the result, we find that the performance peaks around $s = 20$ in most datasets and the result of $s = 30$ and $s = 50$ has no significant difference. However, in dataset Heart, Diabetes, and NoShow, a smaller $s$ have a even higher predictive performance, suggesting that considering more candidates might not always be the best choice. This ablation study indicates that the optimal $s$ is dataset-specific and can be tuned for improved predictive performance in DynFrs.
>
> Thank you once again for your time and effort in reviewing our submission.
>
> Best regards,
>
> Authors of Submission 9406

---

### Official Review · Reviewer_DnwV · 2024-10-29

**Soundness:** 4
**Presentation:** 3
**Contribution:** 3
**Rating:** 8
**Confidence:** 4

**Summary:**

The paper introduces  DYNFRS, a framework that enables efficient machine unlearning in Random Forest models. The objective is to propose a framework that supports the unlearning of a noticeable amount of data, i.e., above 0.1% of the original training set, without harming the performance of the model. Moreover, the framework should support real-world requests, such as adding data to the training set and querying the model. This is achieved through the three components of the framework: (i) the One-Class Constraint subsampling (OCC) that restricts each data sample to appear only in a subset of trees within the forest, reducing the influence of each sample on the model and thereby simplifying the unlearning process; (ii) the Lazy Tag Strategy (LZY) that marks only the nodes affected by data deletion or addition, allowing the algorithm to defer modifications until a relevant query necessitates them; (iii) the adoption of Extremely Randomized Trees (ERTs) as base learners in the forest that enable resilience to train data variations due to their inherent randomness in split selection. The time complexity of the data deletion, data addition, and querying of the model is formally proven, as well as the exactness of the unlearning procedure.

On nine binary classification datasets, DYNFRS is evaluated against other state-of-the-art machine unlearning approaches (DaRE, HedgeCut, and OnlineBoosting). The experimental results show that forests trained with DYNFRS present predictive performance that is at least equal to the ones of the other frameworks. Moreover, DYNFRS outperforms the other methods in sequential and batch unlearning tasks, demonstrating speedups up to 10^6x over traditional retraining and low growth of the unlearning runtime with a high number of requests with respect to the competitors. DYNFRS also supports requests in dynamic environments where data deletion, addition, and querying requests interleave, achieving a very low latency for modifications and making it practical for real-world applications.

**Strengths:**

This is a good paper that presents a significant contribution to the unlearning field.

The novelty of this work stands in the proposed framework. Even though DYNFRS includes some techniques already presented in DaRE, in particular the random splits and the storage of the updated statistics of the nodes involved in the unlearning phase, the combination of these two techniques with constrained subsampling (OCC) and the tagging strategy (LZY) is novel and it has never been explored in the literature. Moreover, the framework supports not only data unlearning but also data addition and can handle a mix of requests in sequence, capabilities that are not supported by the considered state-of-the-art baselines.

The significance of the framework is highlighted by its performance with respect to the baselines. In sequential unlearning and batch unlearning, DYNFRS presents a speedup of up to 10^5 times with respect to the baselines. Fast learning and unlearning make the framework directly usable in real-world applications. However, more details about the experimental methodology need to be assessed to understand if the comparison with the baselines is completely fair (see the Weaknesses Section).

Finally, the presentation of the algorithms is clear, but the presentation throughout the paper can be improved in general (see the Weaknesses Section). It is appreciable that the authors have formally proven the complexity of the data unlearning, data addition and model querying procedures.

**Weaknesses:**

Even though the proposal of the paper is good, the paper presents some weaknesses and unclear points that the authors should address.

**Noteworthy weaknesses**

The first point regards the presentation of the experimental methodology, which is not deepened and not clear. In Section 5.1.1., it is written that `For all baseline models, we adhere to the instructions provided in the original papers and use the same parameter settings.`. Thus, it is not guaranteed that the baselines are using the same hyperparameter values of the forests trained by DYNFRS, like the number of trees and maximum depth, since, for example, DaRE expects to tune these parameter values. While this is acceptable to compare the accuracy of the best performing models trained by different frameworks, the models should all have the same hyperparameter values when comparing the training and unlearning time cost since the complexity of training and unlearning depends on the number of trees and maximum depth of the models. I invite the authors to clarify this point and provide experimental results related to a fairer comparison. Moreover, it is unclear how often the experiments are repeated to obtain the error bars and standard deviations shown in tables and figures and what is different among each repetition.

Second, it would be appreciated if the authors could provide the space complexity of a DYNFRS forest, as done in the DaRE paper for the DaRE forests. Indeed, it is not clear if the gain in time efficiency obtained by DYNFRS comes at the cost of a higher memory consumption. The suggested analysis would allow the authors to clarify this point. Moreover, an experimental comparison of the memory consumption of DYNFRS during the training phase with the baselines and an example of the memory consumption of a DYNFRS forest during a mix of requests of different types may highlight the practicality of the proposed framework.

Finally, the framework is tested only on binary classification tasks. It is not clear from the paper if the framework supports also multiclass classification tasks. The authors should provide clarifications about this point and, if the framework supports multiclass classification tasks, replicate the presented experiments including some multiclass datasets. If the framework does not support multiclass classification tasks, it is not a big limitation, but the authors should clearly state the reason of this limitation.

**Minor points**

The paper contains a lot of typos, especially in the notation of the Section 3. I invite the authors to fix them. Examples of typos are:
- page 3, line 132, $S, <x, y>)$: $<x, y>$ is missing curly brackets.
- page 3, line 134: $A(S \cup <x, y>)$: $<x, y>$ is missing curly brackets.
- In Section 3.2, $(a^*_u, w^*_u)$ and $p$ are not properly introduced.
- page 5, line 4: $D^n$ has never been defined.
- Page 5, line 142: Is the indicator function missing?
- In Section 4.2., the sentence `Unlike OCC(q)’s reducing work across trees and believing in ensemble, LZY relies on tree structures,
dismantling requests into smaller parts and digesting them through time.` does not make any sense and should be rephrased.

**Update after authors' response**
The authors have clarified the weaknesses listed above and the questions below in a satisfactory way, so I have increased my score. I leave the weaknesses written above to keep a trace of the discussion.

**Arguments for the score**

The proposal is novel and significant. Even though the proposed framework combines existing ideas, this combination gives rise to a new solution that is more efficient and effective than state-of-the-art solutions. Moreover, the framework supports operations not supported by other unlearning frameworks in the literature and presents a notable speedup over recent baselines. The experimental results are convincing about the improved efficiency and accuracy of the trained forests with respect to state-of-the-art forests. Thus, I recommend the acceptance of the paper.

**Questions:**

- Do the models trained by each framework in the experimental evaluation have the same hyperparameter values, e.g., number of trees and maximum depth?
- How much memory does DYNFRS consume during training compared to the baselines? How much memory does DYNFRS consume during a stream of mixed requests?
- The tables and figures in the experimental evaluation sections show error bars. However, the description of the experimental methodology makes how these error bars are computed unclear. How many times each experiment is repeated? What differs in each repetition of the experiments?
- Why are multiclass datasets not used in the experimental evaluation? Does DYNFRS also support multiclass classification tasks? If yes, how do models trained with DYNFRS perform in terms of accuracy, unlearning efficiency, and latency on a stream of mixed requests?
- In Section 4.1, the speedup is computed as $N_{occ}/N_{nai}$. Instead, should it be computed as $N_{nai}/N_{occ}$? Moreover, is the speedup achieved by OCC over the naive method $1/q^2$ or  $1/q$? In Section 4.1, both the values are used.

**Details Of Ethics Concerns:**

.

---

> ### Author Response · Authors · 2024-11-21
> **Reply to Reviewer DnwV**
>
> We greatly appreciate the reviewers' detailed and constructive feedback, which has helped us improve the quality and clarity of our manuscript. Here are the responses to the unclear points and weaknesses you care about.
>
> ### **1. Do the models trained by each framework in the experimental evaluation have the same hyperparameter values, e.g., number of trees and maximum depth?**
>
> No. The hyperparameters used in each model are tuned individually on each dataset so that the number of trees and maximum depths might differ. However, DynFrs tends to have more trees and maximum depths than DaRE. Here, we include the elapsed time for DynFrs with DaRE's hyperparameters (we call it DaFrs) to unlearn 1, 10, 100 samples, and 0.1% and 1% of the whole training set.
>
> || Unlearn 1 Sample Time ($\downarrow$,ms) | Unlearn 1 Sample Time ($\downarrow$,ms) | Unlearn 1 Sample Time ($\downarrow$,ms) | Unlearn 10 Samples Time ($\downarrow$,ms) | Unlearn 10 Samples Time ($\downarrow$,ms) | Unlearn 10 Samples Time ($\downarrow$,ms) | Unlearn 100 Samples Time ($\downarrow$,ms) | Unlearn 100 Samples Time ($\downarrow$,ms) | Unlearn 100 Samples Time ($\downarrow$,ms) |
> |---|---|---|---|---|---|---|---|---|---|
> |**Datasets**|DaRE|DaFrs|DynFrs|DaRE|DaFrs|DynFrs|DaRE|DaFrs|DynFrs|
> |**Purchase**|35.0$\pm15$|0.0$\pm0.0$|0.40$\pm0.2$|295$\pm54$|3.80$\pm2.3$|12.3$\pm4.8$|3591$\pm186$|28.0$\pm3.4$|70.4$\pm11$|
> |**Vaccine**|16.0$\pm15$|0.0$\pm0.0$|1.40$\pm0.9$|285$\pm160$|1.60$\pm1.5$|9.20$\pm2.8$|2385$\pm408$|14.6$\pm3.2$|82.6$\pm11$|
> |**Adult**|10.6$\pm5.0$|0.0$\pm0.0$|1.10$\pm1.5$|148$\pm79$|1.20$\pm1.2$|4.80$\pm2.4$|954$\pm158$|13.6$\pm2.2$|32.2$\pm4.0$|
> |**Bank**|33.2$\pm17$|0.0$\pm0.0$|2.40$\pm3.7$|320$\pm108$|1.60$\pm0.8$|7.60$\pm2.7$|3546$\pm302$|25.8$\pm5.6$|78.8$\pm13$|
> |**Heart**|16.8$\pm10$|0.0$\pm0.0$|0.50$\pm0.2$|162$\pm46$|2.80$\pm1.5$|6.80$\pm2.8$|1502$\pm543$|18.4$\pm2.4$|59.3$\pm11$|
> |**Diabetes**|293$\pm168$|5.80$\pm6.2$|7.30$\pm5.4$|2773$\pm877$|53.0$\pm19$|85.5$\pm38$|23833$\pm10330$|430$\pm43$|578$\pm50$|
> |**NoShow**|330$\pm176$|0.0$\pm0.0$|0.30$\pm0.0$|2217$\pm723$|15.6$\pm8.4$|18.2$\pm123$|23856$\pm3978$|165$\pm29.7$|117$\pm10$|
> |**Synthetic**|2265$\pm3523$|0.0$\pm0.0$|2.50$\pm3.7$|92279$\pm42634$|3.00$\pm3.0$|77.3$\pm34$|1073356$\pm420053$|80.4$\pm28$|889$\pm287$|
> |**Higgs**|174$\pm135$|1.20$\pm2.4$|1.60$\pm1.8$|20119$\pm31897$|4.00$\pm3.6$|32.1$\pm8.6$|165122$\pm149092$|84.6$\pm30$|642$\pm317$|
>
> Table: The time to simultaneously unlearn $1$, $10$, and $100$ instances of each model with standard deviation shown in a smaller font. "DaFrs" denotes a DynFrs using DaRE's hyperparameters. "/" means the model takes too long to complete the task.
>
> || Train Time ($\downarrow$,s) | Train Time ($\downarrow$,s) | Train Time ($\downarrow$,s) | 0.1% Unlearning Time ($\downarrow$,s) | 0.1% Unlearning Time ($\downarrow$,s) | 0.1% Unlearning Time ($\downarrow$,s) | 1% Unlearning Time ($\downarrow$,s) | 1% Unlearning Time ($\downarrow$,s) | 1% Unlearning Time ($\downarrow$,s) |
> |---|---|---|---|---|---|---|---|---|---|
> |**Datasets**|DaRE|DaFrs|DynFrs|DaRE|DaFrs|DynFrs|DaRE|DaFrs|DynFrs|
> |**Purchase**|3.10$\pm0.0$|0.16$\pm0.0$|0.38$\pm0.0$|0.35$\pm0.1$|0.00$\pm0.0$|0.01$\pm0.0$|3.39$\pm0.7$|0.03$\pm0.0$|0.07$\pm0.0$|
> |**Vaccine**|4.78$\pm0.0$|0.29$\pm0.0$|1.12$\pm0.0$|0.47$\pm0.1$|0.00$\pm0.0$|0.02$\pm0.0$|5.01$\pm1.0$|0.03$\pm0.0$|0.13$\pm0.0$|
> |**Adult**|5.02$\pm0.1$|0.30$\pm0.0$|0.61$\pm0.0$|0.44$\pm0.3$|0.00$\pm0.0$|0.01$\pm0.0$|3.39$\pm0.6$|0.03$\pm0.0$|0.09$\pm0.0$|
> |**Bank**|8.26$\pm0.2$|0.49$\pm0.0$|1.15$\pm0.0$|1.15$\pm0.3$|0.00$\pm0.0$|0.02$\pm0.0$|14.7$\pm2.3$|0.07$\pm0.0$|0.16$\pm0.0$|
> |**Heart**|12.1$\pm0.2$|0.48$\pm0.0$|1.04$\pm0.0$|0.70$\pm0.2$|0.01$\pm0.0$|0.03$\pm0.0$|8.43$\pm1.2$|0.07$\pm0.0$|0.20$\pm0.0$|
> |**Diabetes**|123$\pm1.0$|7.19$\pm0.0$|8.67$\pm0.0$|23.8$\pm2.7$|0.33$\pm0.0$|0.57$\pm0.1$|258$\pm20$|1.54$\pm0.0$|2.50$\pm0.1$|
> |**NoShow**|65.4$\pm0.4$|3.46$\pm0.0$|3.08$\pm0.0$|18.8$\pm2.7$|0.16$\pm0.0$|0.10$\pm0.0$|268$\pm7.5$|0.67$\pm0.0$|0.56$\pm0.0$|
> |**Synthetic**|1334$\pm6.3$|14.0$\pm0.0$|66.3$\pm0.2$|10790$\pm5348$|0.78$\pm0.2$|5.68$\pm0.2$|/|4.10$\pm0.3$|27.4$\pm0.9$|
> |**Higgs**|10793$\pm48$|187$\pm0.4$|548$\pm1.2$|/|8.57$\pm1.7$|39.2$\pm0.7$|/|40.9$\pm2.6$|201$\pm9.6$|
>
> Table: The training time, and 0.1% and 1% training set unlearning time of each model with standard deviation shown in a smaller font. "DaFrs" denotes a DynFrs using DaRE's hyperparameters. "/" means the model takes too long to complete the task.
>
> As shown in the above tables, DaFrs (DynFrs using DaRE's hyperparameters) has a more splendid efficiency than vanilla DynFrs. The reason is that DynFrs has a larger $T$ (number of trees in the forest) and $d_{\max}$ (maximum depth for each tree) in all datasets except dataset NoShow.

---

> ### Author Response · Authors · 2024-11-21
> **Reply to Reviewer DnwW (cont'd)**
>
> ||DynFrs|DynFrs|DynFrs|DaRE|DaRE|DaRE|
> |---|---|---|---|---|---|---|
> |**Datasets**|$T$|$d_{\max}$|$s$|$T$|$d_{\max}$|$s$|
> |**Purchase**|250|10|30|100|20|25|
> |**Vaccine**|250|20|5|50|20|5|
> |**Adult**|100|20|30|50|20|5|
> |**Bank**|250|25|30|100|20|25|
> |**Heart**|150|15|5|100|10|5|
> |**Diabetes**|250|30|5|250|20|5|
> |**NoShow**|250|20|5|250|20|10|
> |**Synthetic**|150|40|30|50|20|10|
> |**Higgs**|100|30|20|50|20|10|
>
> Table: Hyperparameters used by DynFrs and DaRE.
>
> ### **2. How much memory does DynFrs consume during training compared to the baselines? How much memory does DynFrs consume during a stream of mixed requests?**
>
> The space complexity and memory consumption are discussed in question **1. What are the space complexity and memory consumption of DynFrs?** in public comment **Frequently Asked Questions**; please read it there. Additionally, we tested the maximum resident set size of DynFrs handling a mixed online data stream on Higgs with 100,000 sample additions, 100,000 sample deletions, and 100,000 queries. It turns out that handling the mixed online data stream barely introduces extra space from training as the maximum resident set size is 58327 MB, which is close to the training memory consumption (58263 MB).
>
> ### **3. How many times each experiment is repeated? What differs in each repetition of the experiments?**
>
> All experiments in this paper have been repeated 5 times, and all reported statistics (e.g., average, minimum, maximum, and standard deviation) are calculated directly from these 5 outcomes. The random seeds are different each time, and we use the `srand(time(NULL))` settings, meaning that the random seed changes with system time. We mention the implementation details in the appendix and will consider moving this part with extra details to the main text to avoid confusion. Thank you for pointing this out!
>
> ### **4. Why are multiclass datasets not used in the experimental evaluation? Does DYNFRS also support multiclass classification tasks?**
>
> DynFrs support multi-class classification. The reason that we did not evaluate DynFrs on multi-class classification in the first place is that it is harder to implement a multi-class forest than a binary-class forest. The other two Random Forest unlearning methods (DaRE and HedgeCut) do not have implementation for multi-class classification as well, so it will be hard to find such a baseline. However, we implement a multi-class DynFrs as discussed in question **2. Is DynFrs capable of conquering multi-class classification?** in public comment **Frequently Asked Questions**.
>
> ### **5. In Section 4.1, the speedup is computed as $N_{\text{occ}} / N_{\text{nai}}$. Instead, should it be computed as $N_{\text{nai}} / N_{\text{occ}}$? Moreover, is the speedup achieved by OCC over the naive method $1 / q^2$ or $1 / q$?**
>
> We would like to thank the reviewer again for pointing out this typo! The expected unlearning speedup should be calculated as $\mathbb E[N_{\text{nai}} / N_{\text{occ}}] = 1/q^2$. DynFrs will be $1/q$ times faster than the naive approach when **training**. After being trained, by retraining necessary $\lceil qT \rceil$ trees, the speedup of DynFrs against the naive approach in **unlearning one sample** is $1/q^2$. It means OCC alone brings a $1/q^2$ boost without LZY or ERTs. We will fix our manuscript later.

---

> ### Comment · Reviewer_DnwV · 2024-11-22
> **Thank you for your response**
>
> Thank you for your response. The new results you provided are convincing, and I am very satisfied with the quality of your response. I encourage you to incorporate these new results and discussions into the next version of your paper. Additionally, please address the indicated typos. I will increase my score as a consequence of your high quality response.

---

### Official Review · Reviewer_7Tpx · 2024-11-02

**Soundness:** 2
**Presentation:** 2
**Contribution:** 2
**Rating:** 5
**Confidence:** 2

**Summary:**

This paper focuses on the unlearning of random forests and proposes the DYNFRS framework. DYNFRS is based on extremely randomized trees and includes subsampling and lazy tagging to improve the efficiency of unlearning procedures. This paper also proves the efficiency advantages of DYNFRS in both theoretical and experimental results.

**Strengths:**

1. The motivation and the contributions of this paper are clearly shown.
2. This paper provides both theoretical and experimental results to show the huge efficiency improvements compared with the retrained model.

**Weaknesses:**

1. This paper targets the unlearning in random forests. However, the proposed method relies on extremely randomized trees instead of the general decision trees, which limits the applicability.

2.  The technical part, as well as the pseudo-code in the appendix, are hard to follow. An overall pipeline or workflow can be better used to understand the proposed method.

3. The experiment, especially the performance comparisons, lacks many essential results, and it cannot prove the effectiveness of the proposed methods. Please refer to questions 3, 4, and 5.

**Questions:**

1.  What are the performance comparisons on the sequential unlearning, batch unlearning, and mixed online stream unlearning settings? The current draft only provides efficiency comparisons.

2. How the proposed method handles the changes of optimal split is not clear. This is the key point about the soundness of the proposed method.

3. In the experiment, what are the detailed settings for the unlearning request in each dataset?

4. In the baselines, this paper chooses four methods corresponding to different tree construction approaches. Thus, does it mean such baselines also contain different initialized well-trained models? In addition, does the random forest stand for the retrained model?

5. In the experiment results in Table 2, why does higher prediction performance stand for better unlearning performance? In addition, in the unlearning of a tree-based method, which has more transparent structures, is it a better way to prove the success of unlearning by showing that the unlearned trees have the same structures as the retrained trees?

I am not familiar with the extremely randomized trees, so I might change my assessment based on other reviewers' comments.

---

> ### Author Response · Authors · 2024-11-21
> **Reply to Reviewer 7Tpx**
>
> We appreciate the reviewers' constructive feedback and the opportunity to address their thoughtful questions, ensuring our work achieves its full potential. Here is more background information that is helpful in answering your questions.
>
> ### **0. Clarifications on metrics**
>
> There are three different metrics to evaluate an unlearning method --- predictive performance (how well can the model predict?), unlearning exactness (does the model forget everything we ask it to forget?), and unlearning efficiency (how fast is the forgetting process?).
>
> **Predictive performance** evaluates how accurate the model is on classification or regression tasks, and it has nothing to do with unlearning exactness and unlearning efficiency. The goal of evaluating predictive performance is to check whether the framework sacrifices accuracy for unlearning efficiency.
>
> **Unlearning exactness** evaluates the behavior of the model when processing unlearning requests. An ideal unlearning framework should have the same behavior as the one being retrained from scratch. Exactness ensures that the framework forgets everything about the samples deleted. Unlearning exactness is essential to privacy-sensitive fields where tree-based methods are employed. A detailed definition can be found in Section 3.1 of the manuscript.
>
> **Unlearning efficiency** indicates how fast the model unlearns one or a batch of samples. Retraining the model ensures good predictive performance and excellent unlearning exactness, but it is time-consuming and resource-demanding, which contradicts the goal of minimizing the time and computational resources required to unlearn samples. In reality, we cannot wait for a long time to just unlearn one single sample, so high unlearning efficiency means better real-world applicability.
>
> The goal of unlearning frameworks is to: (A) achieve a similar (comparative) predictive performance with the retraining model before and after unlearning (achieve exact unlearning); (B) optimize unlearning efficiency to reduce time and resource consumption;
>
> In our manuscript, we check goal (A) in Section 5.2 Table 2, showcasing that DynFrs does not sacrifice accuracy for unlearning efficiency, and in Theorem 1 and Figure 3, demonstrating that DynFrs achieves exact unlearning theoretically and empirically. Moreover, Sections 5.3 and 5.4 present DynFrs's outstanding unlearning efficiency for pursuing goal (B).
>
> Here are the responses to the questions you raised:
>
> ### **1. What are the performance comparisons on the sequential unlearning, batch unlearning, and mixed online stream unlearning settings? The current draft only provides efficiency comparisons.**
>
> We argue that performance comparisons on all unlearning settings are not necessary as DynFrs has been proven to have unlearning exactness. If the unlearning model has a higher accuracy than the retraining model, it might indicate that the unlearning model takes advantage of the deleted samples to achieve better accuracy, which is harmful in privacy-sensitive settings. Recall that goal (A) states that the model should have similar (actually "equivalent" for DynFrs) behavior as the retraining model after completing the unlearning task. And we have proven theoretically (Theorem 1) and empirically (Figure 3) that DynFrs enjoys unlearning exactness, so there is no worry about its predictive performance after unlearning. Therefore, our experiments focus on goal (B) --- evaluating DynFrs's unlearning efficiency. Additionally, both baselines DaRE and HedgeCut focus on demonstrating the model's unlearning efficiency. Finally, we would like to thank you for raising such concern, and we will adjust the description in Section 5 to demonstrate the purpose of each experiment.
>
> ### **2. How the proposed method handles the changes of optimal split is not clear. This is the key point about the soundness of the proposed method.**
>
> Without the lazy tag, when a change in optimal split happens in node $u$, we simply retrain the whole subtree $u$. But with the presence of lazy tags, we destroy that subtree (delete every node other than $u$) and place a lazy tag on node $u$ to indicate this node is not a normal leaf and should be grown when a query visits this leaf. When a query does visit the said tagged node $u$, we find the optimal split for this node from all stored split candidates, grow two leaves with lazy tags from $u$, and remove $u$'s lazy tag. Then, the query will only visit one of its leaves, and the growing progress continues recursively until a tree path from $u$ to a leaf is constructed. A detailed growing process can be found in Section 4.2, and how a new optimal split is found is in Section 4.3.

---

> ### Author Response · Authors · 2024-11-21
> **Reply to Reviewer 7Tpx (cont'd)**
>
> ### **3. In the experiment, what are the detailed settings for the unlearning request in each dataset?**
>
> For each dataset, the unlearning settings are the same. For sequential unlearning, we feed the model with training samples one by one in random order. For batch unlearning of size $|S|$, we randomly pick $|S|$ different training samples to form the unlearning batch and feed the batch to the model. We will add this information to our manuscript later. Moreover, for batch unlearning, we explicitly write the batch size $|S|$ in the plot title (Figure 5) for all experiments.
>
> ### **4. In the baselines, this paper chooses four methods corresponding to different tree construction approaches. Thus, does it mean such baselines also contain different initialized well-trained models? In addition, does the random forest stand for the retrained model?**
>
> There is no pre-training in our experiments. For all baselines (DaRE, HedgeCut, OnlineBoosting, and Random Forest Classifier implemented by scikit-learn), we trained them thoroughly on our server for experiments. Also, DynFrs does not have back-bone or pre-trained components, so we train repeatedly for each experiment. For the second question, "Random Forest" in Table 2 stands for the vanilla Random Forest Classifier implemented in scikit-learn, and it does not contain specializations for unlearning (Yes, it is the retraining model). We might change "Random Forest" to "Vanilla" in our manuscript for less confusion.
>
> ### **5. In the experiment results in Table 2, why does higher prediction performance stand for better unlearning performance? In addition, in the unlearning of a tree-based method, which has more transparent structures, is it a better way to prove the success of unlearning by showing that the unlearned trees have the same structures as the retrained trees?**
>
> It is an interesting question.
> First of all, higher prediction performance does not stand for better unlearning performance.
> As mentioned in **Clarifications on metrics**, predictive performance does not imply anything about unlearning exactness and unlearning efficiency, so higher prediction performance does not stand for better unlearning exactness (or success of unlearning as you mentioned) nor unlearning efficiency. Table 2 only guarantees that DynFrs does not sacrifice accuracy for unlearning efficiency, as the accuracy is measured right after training and before unlearning.
>
> For the second question, we think you are on the right track for checking similar tree structures. However, tree structures can be quite different even for the same building algorithm and same hyperparamters but different random seeds, as Random Forests embrace a fair amount of randomness. So, the idea is to see whether the unlearned model has a similar behavior as the retrained model (i.e., the distribution of their predictions are the same), which is the definition of unlearning exactness (Section 3.1), and your idea is similar to that of goal (A) which has been examined theoretically (Theorem 1) and empirically (Figure 3). Thank you again for raising these points, and we will adjust our manuscript accordingly to avoid further confusion.
>
> ### **6. Addition reply to Weakness 1: However, the proposed method relies on Extremely Randomized Trees instead of the general decision trees, which limits the applicability.**
>
> DynFrs can be applied to other trees, such as the vanilla Random Forest and other kinds of trees, by simply substituting the Extremely Randomized Trees with other base learners since OCC and LZY have little requirement for the tree structure. But we choose Extremely Randomized Trees here for its limited dependence on training samples, which is crucial for achieving better unlearning efficiency.

---

> ### Author Response · Authors · 2024-11-25
>
> Dear Reviewer 7Tpx,
>
> We would like to express our gratitude for your constructive suggestions and thoughtful reviews, which have proven invaluable in enhancing the quality of our paper. As a follow-up to our rebuttal, we would like to kindly remind you that the deadline for discussion closure is rapidly approaching.
>
> During this open response period, we aim to engage in discussions, address any further inquiries, and further enhance the overall quality of our paper. We would appreciate it if you could confirm whether you have had the opportunity to review our rebuttal, in which we made concerted efforts to address all of your concerns. It is of utmost importance to us that our responses have been thorough and persuasive.
>
> Should you require any additional information or clarification, please do not hesitate to let us know. Thank you once again for your time and valuable consideration.
>
> Best regards,
>
> Authors of Submission 9406

---

### Official Review · Reviewer_JhHN · 2024-11-04

**Soundness:** 3
**Presentation:** 4
**Contribution:** 3
**Rating:** 8
**Confidence:** 4

**Summary:**

This paper presents a novel unlearning approach for random forests. The authors employ a variant of bootstrapping, named OCC(q), to control the number of trees that each sample is used to train. This bootstrapping technique significantly decreases the number of trees that require unlearning, addressing a gap in existing random forest unlearning methods. Additionally, the authors propose lazy tagging, LZY, to defer re-training of subtrees until needed, enabling batched deletion and speeding up addition and deletion requests. The authors theoretically prove the exactness of their algorithm and analyze its time complexity. They conclude with comprehensive experiments demonstrating their method's superior performance compared to existing approaches.

**Strengths:**

* The method is simple yet effective, yielding an exact unlearning method for random forests.
* The approach reduces required re-training from the source by limiting the number of trees each data point influences.
* The method defers subtree re-computation until queries require that portion, making it suitable for online settings.
* Theoretical proofs demonstrate the algorithm's exactness and time complexity.
* Comprehensive experiments show superior performance compared to baselines, which suffer from numerical value binning.
* The algorithm demonstrates significantly faster performance than baselines.

**Weaknesses:**

* Standard bootstrapping should be included as a baseline for performance comparison. The method involves two key changes: fixing the number of trees each sample is used in and using extremely randomized trees. Including separate comparisons would provide insights towards the impact of each modification.
* Space complexity analysis is missing from the paper. While the authors mention what additional information is stored for each node in Section 4.3, an explicit discussion would be valuable.
* The evaluation lacks testing on regression datasets.
* Line 204: \in -> \subset

**Questions:**

* Can the authors address the points raised in the weakness section above?

---

> ### Author Response · Authors · 2024-11-21
> **Reply to Reviewer JhHN**
>
> We are grateful to the reviewers for their careful reading and valuable suggestions, which have significantly contributed to refining our research. Here are the responses to your points raised in the weakness sections.
>
> ### **1. Standard bootstrapping should be included as a baseline for performance comparison.**
>
> As suggested, we constructed DynFrs with standard bootstrapping instead of OCC subsampling (call it BootFrs) and evaluated its unlearning boost. Here are the results compared with DaRE and ordinary DynFrs:
>
> | | Train Time ($\downarrow$,s) | Train Time ($\downarrow$,s) | Train Time ($\downarrow$,s) | Unlearn Boost ($\uparrow$) | Unlearn Boost ($\uparrow$) | Unlearn Boost ($\uparrow$) |
> |---|---|---|---|---|---|---|
> | **Datasets** | DaRE | BootFrs | DynFrs | DaRE | BootFrs | DynFrs |
> | **Purchase** | 3.10$\pm0.0$ | 2.94$\pm0.0$ | 0.38$\pm0.0$ | 88.2$\pm6$ | 311$\pm80$ | 4871$\pm311$ |
> | **Vaccine** | 4.78$\pm0.0$ | 12.8$\pm0.0$ | 1.12$\pm0.0$ | 218$\pm28$ | 1314$\pm74$ | 21365$\pm0$ |
> | **Adult** | 5.02$\pm0.1$ | 5.81$\pm0.0$ | 0.61$\pm0.0$ | 467$\pm17$ | 1669$\pm257$ | 22993$\pm801$ |
> | **Bank** | 8.26$\pm0.2$ | 13.7$\pm0.1$ | 1.15$\pm0.0$ | 233$\pm27$ | 1470$\pm231$ | 30257$\pm2600$ |
> | **Heart** | 12.1$\pm0.2$ | 8.36$\pm0.0$ | 1.04$\pm0.0$ | 891$\pm113$ | 1823$\pm578$ | 19729$\pm146$ |
> | **Diabetes** | 123$\pm1.0$ | 99.7$\pm0.1$ | 8.67$\pm0.0$ | 345$\pm38$ | 2074$\pm268$ | 30066$\pm706$ |
> | **NoShow** | 65.4$\pm0.4$ | 30.1$\pm0.1$ | 3.08$\pm0.0$ | 222$\pm45$ | 3552$\pm303$ | 45212$\pm4595$ |
> | **Synthetic** | 1334$\pm6.3$ | 815$\pm0.8$ | 66.3$\pm0.2$ | 271$\pm326$ | 6008$\pm475$ | 141798$\pm2208$ |
> | **Higgs** | 10793$\pm48$ | 5924$\pm39$ | 548$\pm1.2$ | 11708$\pm7719$ | 96150$\pm5303$ | 1498263$\pm26629$ |
>
>
> Table: The training time measured, and unlearning boost of each model with standard deviation shown in a smaller font. "BootFrs" denotes a DynFrs with standard bootstrapping.
>
> Thank you again for your idea of the separate comparison. It provides insight into to what extent ERTs and OCC bring incredible unlearning efficiency to DynFrs.
>
> ### **2. Space complexity analysis is missing from the paper.**
>
> The space complexity and memory consumption are discussed in question **1. What are the space complexity and memory consumption of DynFrs?** in public comment **Frequently Asked Questions**; please read it there.
>
> ### **3. The evaluation lacks testing on regression datasets.**
>
> We have already adopted DynFrs for multi-class classification as discussed in question **2. Is DynFrs capable of conquering multi-class classification?** in public comment **Frequently Asked Questions**.
> We are still working on modifying DynFrs for regression tasks, which other tree-based unlearning methods have never evaluated, and it will be released in a few days.

---

> ### Author Response · Authors · 2024-11-30
> **Reply to Reviewer JhHN (cont'd)**
>
> Here are the results on Regression tasks, which is not implemented and evaluated by any of the existing tree-based unlearning methods. As the first unlearning framework testing on regression tasks, we compare DynFrs's predictive performance (via mean squared error) and unlearning efficiency (via unlearning boost) against the naive retraining method (the scikit-learn Random Forest Regressor implementation, and we label it as "Vanilla" in the following). We used a large and popular dataset Bike (https://archive.ics.uci.edu/dataset/275/) as the regression task.
>
> | **Datasets** | #train | #test | #attr |
> |---|---|---|---|
> | **Bike** | 13903 | 3476 | 16 |
>
> Table: Datasets specifications. (#train: number of training samples; #test: number of testing samples; #attr: number of attributes.)
>
> |  | MSE ($\downarrow$) | MSE ($\downarrow$) | Train Time ($\downarrow$,ms) | Train Time ($\downarrow$,ms) | Unlearn Boost ($\uparrow$) | Unlearn Boost ($\uparrow$) |
> |---|---|---|---|---|---|---|
> | **Datasets** | Vanilla | DynFrs | Vanilla | DynFrs | Vanilla | DynFrs |
> | **Bike** | 3.9042$\pm.071$ | 3.0106$\pm.167$ | 6796$\pm6.5$ | 3412$\pm54$ | 1$\pm0$ | 2368$\pm118$ |
>
> Table: Each model's predictive performance, training time, and unlearning boost with standard deviation shown. "Vanilla" denotes the Random Forest implemented in scikit-learn.
>
> Results show that DynFrs outperforms the vanilla Random Forest in terms of predictive performance and training time. Note that the implementation of DynFrs regressor is rough and poorly optimized due to short time slots. However, DynFrs still shows outstanding unlearning efficiency on regression tasks, as DynFrs achieves an averaged $2368$ unlearning boost in dataset Letter.
>
> We suggest picking $q = 0.5$ for a lower mean squared error in the regression setting. In the experiment, we set the hyperparameters of DynFrs to $T = 100$, $d_{\max} = 15$, and $s = 5$.
>
> Again, We sincerely thank you for your thoughtful and constructive suggestions, which have significantly improved the quality and clarity of our work.

---

### Author Response · Authors · 2024-11-21
**Frequently Asked Questions**

### **1. What are the space complexity and memory consumption of DynFrs?**

The space complexity of DynFrs is $\mathcal O(qTn + Tvps)$ where $T$ is the number of trees in the forest, $n$ is the number of samples in the training sets, $q$ is the factor used in OCC, $v$ is the average number of nodes in each tree, $p$ is the number of attributes considered by each node, and $s$ is the number of candidate splits. Since we store training samples on each leaf, and each tree occupies $qn$ samples on average, then the leaf statistics sums up to $\mathcal O(qTn)$. As mentioned in section 4.3, we store an extra $\mathcal O(ps)$ split statistics on each node, so that these split statistics contribute up to $\mathcal O(Tvps)$ space.

We compare the maximum resident space of DynFrs and DaRE for building on the training set. We found that DaRE, which has a space complexity of $\mathcal O(Tn + Tvps)$, has larger memory consumption than DynFrs in all datasets. We use `/usr/bin/time` in Linux to evaluate the maximum resident set size.

|Models|Purchase|Vaccine|Adult|Bank|Heart|Diabetes|NoShow|Synthetic|Higgs|
|---|---|---|---|---|---|---|---|---|---|
|**DaRE**|398.4$\pm1.4$|782.2$\pm1.6$|460.6$\pm2.2$|801.0$\pm1.8$|254.4$\pm1.7$|7148$\pm39$|3792$\pm19$|8526$\pm65$|60528$\pm789$|
|**DynFrs**|83.2$\pm1.6$|492.8$\pm1.6$|232.0$\pm0$|300.0$\pm0$|252.2$\pm1.6$|2512$\pm7.6$|761.6$\pm2.0$|5827$\pm7.8$|58263$\pm52$|

Table: The maximum resident set size ($\downarrow$) of each model measured in megabytes with standard deviation shown.

We totally agree that space is another important factor apart from time. Thank you all for bringing this out. We will add this analysis and result to the appendix in our manuscript.

### **2. Is DynFrs capable of conquering multi-class classification?**

DynFrs is capable of handling multi-class classification tasks since OCC and LZY do not affect the functionality of the forest. We implemented an Extremely Randomized Tree dealing with multiple label classes for DynFrs and tested DynFrs's prediction performance and unlearning boost on 3 datasets. Unfortunately, existing random forest unlearning methods (DaRE and HedgeCut) have no implementation for multi-class classification, so we only include the Random Forest Classifier implementation (we call it Vanilla in the following) in scikit-learn as our baseline. We use datasets Optical (https://archive.ics.uci.edu/dataset/80/), Pen (https://archive.ics.uci.edu/dataset/81/), and Letter (https://archive.ics.uci.edu/dataset/59/).

|Datasets|#train|#test|#attr|#class|
|---|---|---|---|---|
|**Optical**|3823|1797|64|10|
|**Pen**|7494|3498|16|10|
|**Letter**|15000|5000|16|26|

Table: Datasets specifications. (#train: number of training samples; #test: number of testing samples; #attr: number of attributes; #class: number of label classes.)

|  | Accuracy ($\uparrow$) | Accuracy ($\uparrow$) | Train Time ($\downarrow$,ms) | Train Time ($\downarrow$,ms) | Unlearn Boost ($\uparrow$) | Unlearn Boost ($\uparrow$) |
|---|---|---|---|---|---|---|
|**Datasets**|Vanilla|DynFrs|Vanilla|DynFrs|Vanilla|DynFrs|
|**Optical**|.9694$\pm.002$|.9707$\pm.002$|585.4$\pm1.9$|445.4$\pm2.7$|1$\pm0$|3823$\pm0$|
|**Pen**|.9649$\pm.002$|.9696$\pm.002$|996.2$\pm1.7$|813.2$\pm8.0$|1$\pm0$|7494$\pm0$|
|**Letter**|.9603$\pm.002$|.9624$\pm.001$|1666$\pm7.6$|1530$\pm20$|1$\pm0$|14142$\pm917$|

Table: Each model's predictive performance, training time, and unlearning boost with standard deviation are shown in a smaller font. "Vanilla" denotes the Random Forest implemented in scikit-learn.

Results show that DynFrs outperforms the vanilla Random Forest in terms of predictive performance and training time for all datasets. Note that the implementation of multi-class ERTs in DynFrs is rough and poorly optimized due to short time slots. Multi-class forests are much tougher to implement and optimize compared to binary-class forests, and that may be why existing RF unlearning methods (DaRE and HedgeCut) are not evaluated on multi-class classification datasets. However, DynFrs still shows splendid unlearning efficiency on these datasets. In dataset Optical and Pen, DynFrs finishes unlearning all training samples before the naive retraining method unlearns 1 sample, and DynFrs achieves a 14142 unlearning boost in dataset Letter.

In the multi-class classification setting, we suggest picking $q = 0.5$ for OCC, because the rise in class number leads to the drop of sample size to each class (roughly $n / c$, where $n$ is the number of training samples and $c$ is the number of classes) and rising $q$ enable each tree is accessible to more samples belonging to a specific class and lead to better predictive performance eventually. For all datasets, we set the hyperparameters of DynFrs to $T = 100$, $d_{\max} = 20$, and $s = 1$.

As all of you mentioned, it is important to demonstrate DynFrs's performance on multi-class classification tasks. We will include this part in the appendix to improve the manuscript.

---

### Author Response · Authors · 2024-12-03
**Sincere Thanks to All Reviewers**

Dear reviewers,

As the rebuttal phase draws to a close, we would like to express our heartfelt gratitude for the time and effort you have dedicated to reviewing our submission. Your insightful feedback and constructive comments have been invaluable in refining and strengthening the work.

We deeply appreciate the expertise and attention you brought to this process, and we hope the additional clarifications and revisions provided during the rebuttal phase have addressed your concerns effectively.

Thank you once again for your contribution to this rigorous and collaborative review process.

Warm regards,

Authors of Submission 9406

---

### Meta-Review · Area_Chair_oSUC · 2024-12-24

**Metareview:**

Summary

The paper introduces an efficient approach to enable machine unlearning in random forests, focusing on speed and maintaining predictive accuracy. It uses a subsampling method, OCC(q), which limits each sample's presence to a subset of trees, simplifying the removal or addition of samples. Additionally, a lazy tag strategy defers the retraining of subtrees until necessary, allowing batched deletions and additions while minimizing computational overhead. The method achieves significant efficiency improvements—up to 1,500,000 times faster—compared to naive retraining, with theoretical guarantees for exact unlearning and time complexity analysis. The efficiency is demonstrated empirically with fast unlearning along with competitive predictive accuracy

Strengths

The paper's strengths lie in its simplicity and effectiveness, achieving substantial efficiency improvements while maintaining predictive accuracy. The method minimizes retraining efforts by limiting the number of trees affected by any update and deferring subtree retraining until necessary, making it particularly suitable for online settings and addition beyond unlearning. The authors provide clear and reasonable motivation, supported by both theoretical proofs and time complexity analysis. Empirical results validate the approach, demonstrating significant efficiency gains.

Weaknesses

The paper has several weaknesses but they were (most) partially addressed in the rebuttal and revisions. Initially, it lacked additional comparison experiments, such as evaluations on multi-class datasets and ablation studies, though these were added later. Space complexity concerns were also addressed in the rebuttal. Presentation clarity was a noted issue, which was reportedly improved in the revisions. However, the paper lacks essential empirical comparisons with other unlearning methods. Additionally, the approach relies on extremely randomized trees rather than general decision trees, raising concerns about its generalizability; while the authors claim any tree type can be used, it remains unclear how performance would compare. The evaluation metrics also require clarification: the paper suggests that post-unlearning performance should match retraining, but the rebuttal's explanation is ambiguous about whether this comparison is with retraining performance or the model's performance prior to unlearning. This ambiguity needs further resolution in future iterations.

Recommendation

The paper is recommended as a poster because, while some reviewers noted that the framework's main components are based on existing methods with minor alterations, the combination of these techniques is novel and enables efficient data addition and sequence handling in unlearning. This combination results in significantly superior performance in efficiency, demonstrating that novel integrations of established methods can yield substantial benefits in the unlearning domain. Despite the limited novelty, the practical efficiency, importance of the problem, and potential applications beyond unlearning justify its acceptance as a poster presentation.

**Additional Comments On Reviewer Discussion:**

During the rebuttal and discussion phase, opinions shifted slightly toward favouring acceptance. Reviewer JhHN, who gave 8, had their questions addressed adequately. Reviewer DnwV increased their score to 8 after the rebuttal and supported acceptance as a poster, finding the authors' clarifications satisfactory. Reviewer 7TPx maintained a score of 5 with low confidence (2) and did not participate further in discussions. While some questions remain, particularly around the empirical applicability of the method to tree types beyond extremely randomized trees, clarifications on metrics and experimental details were largely satisfactory. Reviewer THi8 also gave 5 but did not raise additional concerns about novelty; their concerns about ablation studies and presentation clarity were addressed in the rebuttal and revisions. Overall, the discussion leaned toward acceptance as a poster, given the efficiency and practical value of the work.

---

### Decision · Program_Chairs · 2025-01-22

Accept (Poster)